# ProteiNexus: Illuminating Protein Pathways through Structural Pre-training

## Abstract

Protein representation learning has emerged as a powerful tool for various biological tasks. Language models derived from protein sequences represent the predominant trend in many current approaches. However, recent advances reveal that protein sequences alone cannot fully encapsulate the abundant information contained within protein structures, critical for understanding protein function and aiding innovative protein design. In this study, we present ProteiNexus, an innovative approach, effectively integrating protein structure learning with numerous downstream tasks. We propose a structural encoding mechanism adept at capturing fine-grained distance details and spatial positioning. By implementing a robust pre-training strategy and fine-tuning with lightweight decoders designed for specific downstream tasks, our model exhibits outstanding performance, establishing new benchmarks across a range of tasks. The code and models could be found at github repos [1].

## 1 Introduction

Proteins fulfill a myriad of biological roles within organisms, spanning from enzyme catalysis and signal transduction to gene regulation. These biological functions are crucially correlated with the three-dimensional architecture of proteins (Pazos & Sternberg, 2004; Pal & Eisenberg, 2005). For instance, antibodies (such as SARS-CoV-2 (Zhu et al., 2022)), which are integral components of the immune system, initiate a precise immune response against foreign incursions by interacting with antigens present on pathogen surfaces. The specificity and affinity of these interactions hinge on the structure and binding mode of both antibodies and antigens. A deep understanding of protein structures, the interpretation of protein-protein interactions, and the illumination of their respective functions and regulatory mechanisms are fundamental for achieving accurate protein design and precise understanding (Huang et al., 2016).

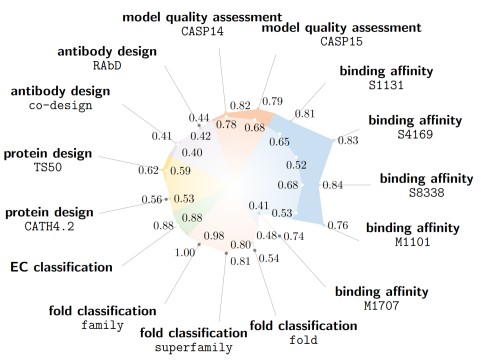

Figure 1: Comparison of results between ProteiNexus and state-of-the-art methods.

Enhancing our understanding of proteins through effective representation learning is paramount for in-depth research. The recent surge in deep learning advancements, especially those related to self-supervised learning, instigates the advent of supremely effective algorithms across myriad tasks within bioinformatics. The advent of high-throughput sequencing leads to an exponential augmentation in protein sequences (Consortium, 2019), motivating the transfer of techniques from large language models (LLMs) such as Transformers (Vaswani et al., 2017) and BERT (Devlin et al., 2018) to protein sequence representation learning, otherwise known as protein language models (pLMs). These sequence-based approaches for protein representation learning triumph in various tasks including function prediction (Nallapareddy et al., 2023; Littmann et al., 2021), protein structure prediction (Rao et al., 2020; Weißenow et al., 2022; Lin et al., 2023), and protein design (Verkuil

---

[1]Upon acceptance of this paper, our codes and models will be made publicly available

et al., 2022; Hie et al., 2022; Ferruz et al., 2022). In parallel, researchers gradually recognize the significance of protein structure and introduce graph-based representations of protein structures (Jing et al., 2021; Somnath et al., 2021; Aykent & Xia, 2022; Li et al., 2022). While this propels the field of protein representation learning forward, it bears its restrictions. Predominantly, graph-based representations struggle to preserve fine-grained atom information effectively. Moreover, they tend to accentuate interactions among neighboring residues while often disregarding the influence of long-range interactions. This limitation becomes particularly pronounced when modeling protein-protein interactions in practical applications. For instance, some specific protein families, like G-protein-coupled receptors (GPCRs), exhibit varying structures when interacting with different ligands, despite sharing identical amino acid sequence (Hilger et al., 2018). Consequently, relying solely on local structural information often results in modeling failures.

Furthermore, most researches focuses on devising robust protein structure encoders, these encoders are tailor-made for particular tasks, thus encountering challenges in maintaining consistently superior performance across a comprehensive array of tasks. To surmount these obstacles, one promising strategy involves the enhancement of performance through pre-training on extensive datasets, contingent upon obtaining effective structural representations (Hermosilla & Ropinski, 2022; Zhou et al., 2023). However, self-supervised learning of three-dimensional protein structures posits inherent complexities. Among prevalent pre-training frameworks, contrastive learning garners notable attention (Hermosilla & Ropinski, 2022; Zhang et al., 2023b). Additionally, other effective strategies include denoising corrupted distance matrices (Zhou et al., 2023) and predicting residual dihedral angles (Chen et al., 2023).

To address these challenges, we present ProteiNexus, a pre-trained model centered on protein structure. ProteiNexus initiates its training regimen with self-supervised learning on extant protein structure data, followed by fine-tuning on an array of downstream tasks including model quality assessment, binding affinity prediction, folding classification, enzyme-catalyzed reaction classification, protein design, and antibody design. We utilize a robust encoder to capture protein distance information and the spatial relative positions of residues, enabling the model to understand representations of interactions learned from pair relationships – affording a more exhaustive understanding of protein complex. Additionally, we amalgamate structural information at both the atom and residue levels, thereby bolstering the model's performance. For added robustness and diversity, we integrate a hybrid masking strategy and mixed-noise strategy. Working in tandem, these strategies empower the model to learn the diversity of protein information more effectively, culminating in exemplary performance across varied tasks.

Our primary contributions can be summarized as follows:

- We present a groundbreaking universal protein pre-training model, adept at seamlessly incorporating both protein sequence and structural information.
- We implement a simple, yet potent, architecture to capture structural information comprehensively. Our model is substantiated through numerous experiments, demonstrating its effectiveness and setting new standards across a diverse range of downstream tasks.

## 2   RELATED WORKS

Protein representation learning is a fundamental challenge in the fields of bioinformatics, aiming to find an effective way to describe the structure and function of proteins. This field can be divided into two major approaches: sequence-based and structure-based methods.

**Protein Sequence Representation Learning.**   Sequence-based protein representation learning is primarily inspired by methods uesd for modeling natural language sequences. Typical pre-training objectives explored in existing methods include next residue prediction, masked language modeling (MLM) and contrastive predictive coding (CPC). There are different masking strategies in masked language modeling (MLM) such as random residue masking (Rao et al., 2021; Rives et al., 2021), pair residue masking (He et al., 2021a), motif or subsequence mask (Wu et al., 2022).

**Protein Structure Representation Learning.**   Protein structure provides direct and valuable information, some approaches (Zheng et al., 2023a) attempt to enhance their performance by fine-tuning parameters of sequence-based pre-trained models and introducing structure-aware modules.

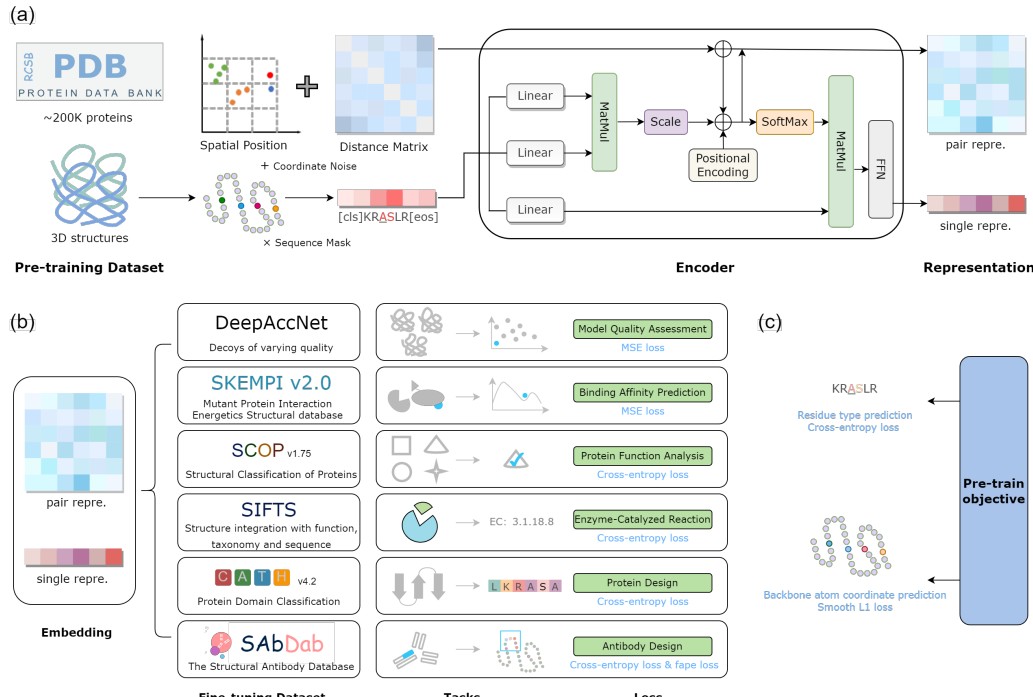

Figure 2: **Method Overview.** (a) Pre-training stage - About 200k proteins from RCSB protein structure database are used to learn protein representation. (b) Fine-tuning stage - Extracting effective representations from pre-training for predictions using lightweight task layers. (c) The objective of pre-training.

Contrastive learning (Hermosilla & Ropinski, 2022; Zhang et al., 2023b) is a highly popular pre-training method designed to learn structural representations by maximizing distance metrics between different protein structures as the training objective. Additionally, there are methods that transfer protein structures into distance matrices and attempt to denoise noisy distance matrices while simultaneously predicting the types of corresponding residue types. These approaches undergo pre-training on large-scale datasets to improve the quality and generalizability of the representations.

## 3 METHODS

### 3.1 PROTEIN REPRESENTATION

Given a protein $\mathcal{P}$ as input, we employ a transformer based model to learn its sequential representation ($\mathbf{s} \in \mathbb{R}^{n \times d_s}$, where n is the number of residues and $d_s$ is single feature dimension) and structural representation ($\mathbf{z} \in \mathbb{R}^{n \times n \times d_z}$, where $d_z$ is the pair feature dimension). Our objective is to capture effective representations of protein sequence and structure through a trainable parameterized model. These representations can be fine-tuned for accurate predictions across a wide range of downstream tasks.

### 3.1.1 ENCODER

Our pretrained model requires two types of inputs: residue type and residue coordinates. Similar to natural language, we represent protein sequence as a sequence of discrete tokens and learn the initial representation $\mathbf{s}^{(0)}$ through a linear layer. To more effectively encode protein structure information and maintain rotation and translation invariance, we employ three distinct encoding methods: Spatial Position Encoding (SPE), Distance Encoding, and Relative Position Encoding (RPE). These methods together constitute the initial pair representation $z^{(0)} = z_{spe} + z_{distance} + z_{rpe}$. These approaches are simpler in nature yet highly effective, allowing for the better preservation of three-dimensional structural information.

**Spatial Position Encoding.** SPE is a method of encoding that is employed to capture the spatial relationships between residues. This encoding technique remains invariant under global rotation and translation. Using the C$\alpha$ atom as the coordinate origin, we establish a local Cartesian coordinate system for each residue through the Schmidt orthogonalization, denoted as the local frame $\mathcal{O}_i$. Subsequently, we project the C$\alpha$ atom of the $j$-th residue onto the local frame $\mathcal{O}_i$, and employ the resulting 3D local coordinates as the spatial position representation. Lastly, we partition the continuous coordinates into bins of equal width and transform each bin into an embedding, which is then utilized as the spatial position encoding, referred to as $z_{spe}$.

**Distance Encoding.** To capture finer-grained structural information, we introduce atom-level distance in this stage, employing the "distance tokenizer" method to efficiently encode protein structural data. Additionally, we establish an alignment mechanism from the atom-level to the residue-level, initializing the distance information as $z_{distance}$. For further details, please refer to the Appendix B.1.

**Relative Position Encoding.** To enrich the network with information about the positional context of residues within the sequence, we introduce relative positional encoding (referred as $z_{rpe}$) into the initial pair representations. Specifically, we employ a one-hot encoding scheme to represent the relative distance between position $i$ and position $j$ in the sequence as a vector. This encoding strategy is restricted to distances less than a predefined threshold, ensuring the effective capture of significant relative positional relationships.

### 3.1.2 BACKBONE NETWORK

Recently, numerous research endeavors in the field of protein structure representation have embraced network architectures based on Graph Neural Networks (GNNs). GNN-based methods have demonstrated remarkable performance in capturing local structural patterns, but challenges persist when dealing with protein complexes. For protein complexes, long-range relationships between residues continue to influence folding configurations and interaction modes to a certain extent. To better capture the global features and interactions of protein structures, we have opted for the transformer architecture as the backbone of our network. This decision is grounded in the inherent self-attention mechanism of the transformer, which enables computations across the entire protein sequence. This capability effectively captures associations between distant residues, thus elevating the precision of structural analysis and prediction. Furthermore, we have introduced a communication mechanism between sequence and structural information, enhancing the model's ability to integrate and exploit insights from both dimensions, resulting in improved prediction outcomes.

The transformer architecture is constructed with stacked layers of transformers, taking initialized single representations as input. Each individual transformer layer is comprised of two primary elements: a self-attention module and a feed-forward network. Updating the single representation in the $l$-th layer is achieved as follows:

$$\text{Attention}(\boldsymbol{Q}_i^{l,h}, \boldsymbol{K}_i^{l,h}, \boldsymbol{V}_i^{l,h}) = \sum_j \text{softmax}\left(\frac{\boldsymbol{Q}_i^{l,h}(\boldsymbol{K}_j^{l,h})^T}{\sqrt{d_k}} + z_{ij}^{l-1,h}\right)\boldsymbol{V}_j^{l,h} \tag{1}$$

where $Q_i^{l,h}$, $K_i^{l,h}$, and $V_i^{l,h}$ correspond to the Query, Key, and Value for the $i$-th residue, in the $l$-th layer and the $h$-th head, $h \in \{1, 2, \ldots, H\}$, H is the number of attention heads, $d_k$ represents the dimension of the Key, and $z_{ij}^{l-1,h}$ denotes the pair representation for the $ij$-th pair in the $l-1$-th layer and the $h$-th head. Furthermore, we utilize the attention weights obtained from the self-attention mechanism to update the pair representations as follow:

$$z_{ij}^{l,h} = z_{ij}^{l-1,h} + \text{Concat}_h\left(\frac{\boldsymbol{Q}_i^{l,h}(\boldsymbol{K}_j^{l,h})^T}{\sqrt{d_k}}\right) \tag{2}$$

### 3.2 PRE-TRAINING

Our training data is derived from the Protein Data Bank (PDB) database, encompassing all protein structure data released up until May 1st, 2023. We employ two self-supervised tasks aimed at

learning universal representations from vast protein structure data. Similar to the field of natural language processing, we adopt a masking strategy, wherein the prediction of masked residues is employed to establish the single representation of proteins. We randomly select a portion of residues along the entire sequence length with varying probabilities for masking and prediction. Due to the interplay between single and pair representations, masked residues can be efficiently reconstructed through structural cues. Consequently, we introduce Gaussian noise into the corresponding pair representations aligned with masked residues to enhance the model's robustness. Moreover, we encourage the model to recover authentic atom-level coordinate from noise-induced residue-level pair representations.

### 3.3 Fine-Tuning on Downstream Tasks

To enhance the model's adaptability to specific tasks, we incorporate lightweight task heads upon the pre-trained model and fine-tune the parameters for downstream tasks. For specific model architectures, please refer to the Appendix C.

## 4 Experiments

In order to verify the effectiveness of our proposed pre-training model, we conduct experiments on several downstream tasks. The implementation details and ablation experiments are provided in Appendix D and E, respectively.

### 4.1 Model Quality Assessment

**Datasets.** Our training dataset includes decoys derived from 7992 unique native protein structures, obtained from DeepAccNet. In the end, we have a collection of 39057 structures in our training dataset, with a fraction representing native structures. This dataset is divided into training and validation sets at a 9:1 ratio. To ensure a fair evaluation of the model's capability in identifying native structures, our test set is meticulously curated. It includes targets with experimentally resolved structures from CASP14 and CASP15, paired with their corresponding predicted structures. To ensure diversity and representativeness, we perform a redundancy reduction process on the test set, limiting sequence identity between targets to within 70%. Notably, due to the division of CASP14 target H1044 into multiple domains (e.g., T1031, T1033), our test set does not include H1044 and its corresponding decoys.

**Baselines.** We compare method with 3 recent or established state-of-the-art baselines. DeepAccNet (Hiranuma et al., 2021) is an excellent method for assessing the quality of protein structures. It employs features like distance maps and residue properties, which are processed through 3D convolution to predict the LDDT score for each residue. Furthermore, it refines the decoy's structure based on error estimation. DeepUMQA (Guo et al., 2022) utilizes Ultrafast Shape Recognition (USR) for efficient feature extraction. These features are then fed into a residual neural network to predict the LDDT score. QATEN (Zhang et al., 2023a) incorporates a self-attention mechanism, representing the decoy structure as a graph, allowing it to predict both LDDT and GDT-TS scores simultaneously.

**Results & discussion.** The results are summarized in Table 1, showcasing our method's superior performance across diverse metrics on the CASP14 and CASP15 test datasets in comparison to other methods. Using the released model parameters, we successfully reproduce the results of the three methods listed in the Table 1 on test datasets. Our approach, which focuses on optimizing both local and global structural quality predictions, continues to achieve optimal results even when compared to DeepAccNet and DeepUMQA, which solely emphasize local structural quality assessment. Furthermore, we observe that despite the larger number of decoys in the DeepAccNet dataset, augmenting our training data with decoys of varying degrees of distortion does not significantly enhance the model's capacity to discern structural quality.

### 4.2 Binding Affinity

**Datasets.** We validate our pre-training model on five datasets, namely S1131 (Xiong et al., 2017), S4169 (Rodrigues et al., 2019), S8338, M1101 (Sirin et al., 2016), M1707 (Zhang et al., 2020). These datasets are mainly derived from SKEMPI (Moal & Fernández-Recio*, 2012), SKEMPI 2.0

Table 1: Comparison of Model Quality Assessment on CASP14 and CASP15 datasets.

| Method | CASP14 | | | | | | | | CASP15 | | | | | | | |
| | GDT-TS | | | | LDDT | | | | GDT-TS | | | | LDDT | | | |
| | RMSE ↓ | $\mathcal{P}$ | $\mathcal{S}$ | $\mathcal{K}$ | RMSE ↓ | $\mathcal{P}$ | $\mathcal{S}$ | $\mathcal{K}$ | RMSE ↓ | $\mathcal{P}$ | $\mathcal{S}$ | $\mathcal{K}$ | RMSE ↓ | $\mathcal{P}$ | $\mathcal{S}$ | $\mathcal{K}$ |
|---|---|---|---|---|---|---|---|---|---|---|---|---|---|---|---|---|
| DeepAccNet (Hiranuma et al., 2021) | - | - | - | - | 0.10 | 0.78 | 0.78 | 0.59 | - | - | - | - | 0.16 | 0.68 | 0.68 | 0.50 |
| DeepUMQA (Guo et al., 2022) | - | - | - | - | 0.11 | 0.78 | 0.76 | 0.57 | - | - | - | - | 0.16 | 0.64 | 0.63 | 0.45 |
| QATEN (Zhang et al., 2023a) | 0.20 | 0.61 | 0.61 | 0.44 | 0.14 | 0.59 | 0.62 | 0.47 | 0.21 | 0.67 | 0.59 | 0.50 | 0.22 | 0.54 | 0.59 | 0.42 |
| ProteiNexus | 0.16 | 0.77 | 0.78 | 0.58 | 0.09 | 0.82 | 0.81 | 0.62 | 0.15 | 0.84 | 0.83 | 0.63 | 0.13 | 0.79 | 0.72 | 0.53 |

Jankauskaite et al. (2018) and AB-Bind (Sirin et al., 2016), three datasets widely uesd for protein interaction prediction collated from experimental data. S1131 is an interface non-redundant single-point mutation from the SKEMPI dataset. S4169 filters all single-point mutation from the SKEMPI 2.0 dataset. S8338 contains all mutations in S4169 and their corresponding reverse mutations. M1101, also known as the AB-Bind dataset, consists of all antibody-antigen complexes. The data in M1707 consists exclusively of multi-point mutations. The original protein structure is referred to as the wild type, and the protein structure with partial residue mutation is referred to as the mutant. Due to the lack of the native three-dimensional structure of the mutant, we hypothesize that the mutation effect does not change the backbone structure of the protein.

**Baselines.** We compare method with 6 recent or established state-of-the-art baselines. FoldX (Schymkowitz et al., 2005) employs an empirical force field to predict the impact of mutations on the binding energy of protein complexes. MutaBind2 (Zhang et al., 2020) utilizes a scoring function composed of seven terms to predict changes in binding affinity. TopGBT and Top-NetTree (Wang et al., 2020) combine topology-based approaches with machine learning techniques. GeoPPI (Liu et al., 2020) employs a geometric representation that learns encoded topological features of protein structures to predict protein-protein interaction effects. The ddg predictor (Shan et al., 2022) utilizes an attention-based geometric neural network. By learning the geometric information of mutation pairs within protein structures and using an attention mechanism, it captures crucial interaction features to predict the effects of mutations.

**Results & discussion.** The results are summarized in Table 2. Our model has demonstrates superior performance on datasets involving single-point mutations and antibody-antigen complexes, surpassing the current state-of-the-art benchmarks. This highlights the model's exceptional capability in accurately capturing inter-chain interactions when characterizing complex structures. Our performance on the multi-point mutation dataset M1707 is less than satisfactory. This may be attributed to the gradual accumulation of mutation effects, which could lead to certain structural changes in the mutant type. However, due to the lack of structural data for mutant types, we use wild-type structures as substitutes, resulting in some bias in the data. In the absence of sufficient data on mutant structures, accurately predicting changes in binding affinity will be a key focus of our future improvement efforts.

Table 2: Results of various binding affinity prediction methods on the mutation dataset. [†] and [♭] denotes results taken from Liu et al. (2020) and Shan et al. (2022), respectively. The top two results are highlighted in **bold** and underlined, respectively.

| Method | S1131 | | S4169 | | S8338 | | M1101 | | M1707 | |
| | Rp ↑ | RMSE ↓ | Rp ↑ | RMSE ↓ | Rp ↑ | RMSE ↓ | Rp ↑ | RMSE ↓ | Rp ↑ | RMSE ↓ |
|---|---|---|---|---|---|---|---|---|---|---|
| FoldX (Schymkowitz et al., 2005)[†] | 0.46 | 2.18 | 0.27 | 2.73 | 0.44 | 2.73 | 0.34 | 2.39 | 0.49 | 3.02 |
| MutaBind2 (Zhang et al., 2020)[†] | - | - | - | - | - | - | - | - | 0.72 | 2.25 |
| TopGBT (Wang et al., 2020)[†] | 0.32 | 2.31 | 0.41 | 1.60 | 0.61 | 1.61 | - | - | - | - |
| TopNetTree (Wang et al., 2020)[†] | 0.29 | 2.4 | 0.39 | 1.65 | 0.59 | 1.65 | - | - | - | - |
| GeoPPI (Liu et al., 2020)[†] | 0.58 | 2.01 | 0.52 | 1.48 | 0.68 | 1.49 | 0.53 | 1.81 | 0.74 | 2.21 |
| ddg predictor (Shan et al., 2022)[♭] | 0.65 | - | - | - | - | - | - | - | 0.59 | - |
| ProteiNexus | 0.81 | 1.57 | 0.83 | 0.98 | 0.84 | 1.23 | 0.76 | 2.04 | 0.41 | 3.01 |

## 4.3 FOLD AND ENZYME-CATALYZED REACTION CLASSIFICATION

**Datasets.** The folding classification of proteins reveals the relationship between protein structure and evolution based on the similarity of protein three-dimensional structures. Following prior works, we collect all protein structure data from the SCOP v1.75 database (Murzin et al., 1995) after clus-

Table 3: Results of classification. [‡] denotes results taken from Jie et al. (2017), [♭] denotes results taken from Hermosilla & Ropinski (2022), [†] denotes results taken from Hermosilla et al. (2021), [♮] denotes results taken from Zhang et al. (2023b) and [∗] denotes results taken from Li et al. (2022). **Bold** and underline indicate the top two results obtained under settings w/o pretraining and w/ pretraining, respectively.

| | Method | Fold | | | React |
|---|---|---|---|---|---|
| | | Fold | Sup | Family | |
| | TMalign (Zhang & Skolnick, 2005)[♭] | 34.0 | 65.7 | 97.5 | - |
| | HHSuite (Steinegger et al., 2019)[♭] | 17.5 | 69.2 | 98.6 | 82.6 |
| | PSI-BLAST (Madeira et al., 2022)[‡] | 5.60 | 42.20 | 96.80 | - |
| **w/o pretraining** | DeepSF (Jie et al., 2017)[‡] | 40.95 | 50.71 | 76.18 | - |
| | LSTM (Rao et al., 2019)[†] | 26.0 | 43.0 | 92.0 | 79.9 |
| | mLSTM (Alley et al., 2019)[†] | 23.0 | 38.0 | 87.0 | 72.9 |
| | CNN Shanehsazzadeh et al. (2020)[♮] | 11.3 | 13.4 | 53.4 | 51.7 |
| | GCN (Kipf & Welling, 2017)[†] | 16.8 | 21.3 | 82.8 | 67.3 |
| | 3DCNN (Derevyanko et al., 2018)[†] | 31.6 | 45.4 | 92.5 | 78.8 |
| | GAT (Veličković et al., 2018)[♮] | 12.4 | 16.5 | 72.7 | 55.6 |
| | EdgePool (Diehl, 2019)[†] | 12.9 | 16.3 | 72.5 | 57.9 |
| | GraphQA (Baldassarre et al., 2021)[†] | 23.7 | 32.5 | 84.4 | 60.8 |
| | GVP (Jing et al., 2021)[♮] | 16.0 | 22.5 | 83.8 | 65.5 |
| | DW-GIN (Li et al., 2022)[∗] | 31.8 | 37.3 | 85.2 | 76.7 |
| | IEConv (Hermosilla et al., 2021)[†] | 45.0 | 69.7 | 98.9 | 87.2 |
| | GearNet-Edge-IEConv (Zhang et al., 2023b)[♮] | **48.3** | **70.3** | **99.5** | **85.3** |
| **w/ pretraining** | DeepFRI (Gligorijević et al., 2021)[†] | 15.3 | 20.6 | 73.2 | 63.3 |
| | ESM-1b (Rives et al., 2021)[♮] | 26.8 | 60.1 | 97.8 | 83.1 |
| | ProtBERT-BFD (Elnaggar et al., 2021)[†] | 26.6 | 55.8 | 97.6 | 72.2 |
| | New IEConv (Hermosilla & Ropinski, 2022)[♭] | 50.3 | **80.6** | 99.7 | 88.1 |
| | Multiview Contrast (Zhang et al., 2023b)[♮] | **54.1** | 80.5 | **99.9** | 87.5 |
| | ProteiNexus | 47.6 | 79.7 | 98.0 | **88.4** |

tering with 95% sequence identity. We then follow the data processing method of Jie et al. (2017), remove the redundancy between the training set, validation set, and test set, and demonstrate the performance of our method on three different levels of test sets. Enzymes with catalytic properties are an important component of proteins, and the Enzyme Commission specifies a set of numbering and naming methods for different categories of enzymes, consisting of four digits. We collect proteins annotated with EC numbers from the SIFTS database (Jose et al., 2018) and divide the dataset following Hermosilla et al. (2021).

**Baselines.** In comparison with the classification task, we examine a range of baseline methods with the aim of comprehensively assessing the performance of our model and providing reference for further investigation. Firstly, we employ traditional methods such as TMalign (Zhang & Skolnick, 2005), HHSuite (Steinegger et al., 2019) and PSI-BLAST (Madeira et al., 2022) as baselines, which have widespread applications in protein structure and sequence similarity analysis. Secondly, our focus turns to sequence-based methods, which primarily utilize the amino acid sequence information of proteins for classification: DeepSF (Jie et al., 2017), LSTM (Rao et al., 2019), mLSTM (Alley et al., 2019) and CNN Shanehsazzadeh et al. (2020). Additionally, we also delve into structure-based methods, which center on the three-dimensional structural information of proteins, encompassing factors such as inter-amino acid distances and secondary structures: GCN (Kipf & Welling, 2017), 3DCNN (Derevyanko et al., 2018), GAT (Veličković et al., 2018), EdgePool (Diehl, 2019), GraphQA (Baldassarre et al., 2021), GVP (Jing et al., 2021), DW-GIN (Li et al., 2022), IEConv (Hermosilla et al., 2021), GearNet (Zhang et al., 2023b). Moreover, some methods employ extensive unlabeled data in their model training through pretraining strategies, aiming to enhance the model's feature representation capabilities. For instance, DeepFRI (Gligorijević et al., 2021) leverages information from the protein sequence database Pfam for pretraining, ESM-1b (Rives et al.,

2021) utilizes the UniRef50 dataset, and ProtBERT-BFD (Elnaggar et al., 2021) integrates the BFD database. Additionally, we also consider approaches that incorporate protein structural information, where New IEConv (Hermosilla & Ropinski, 2022) utilizes the PDB database, and Multiview Contrast (Zhang et al., 2023b) combines data from AlphaFoldDB.

**Results & discussion.** As depicted in Table 3, our model's performance aligns comparably with that of other established baselines. In the realm of fold classification, our model demonstrates robust classification accuracy, accurately assigning protein structures to their respective fold categories. This suggests that our approach effectively captures structural patterns and features crucial for fold discrimination. Furthermore, the close proximity of our results to the baseline tasks indicates the competitiveness of our model in this specific task. Moving on to EC classification, our model exhibits a commendable ability to predict EC number accurately. The obtained results substantiate the efficacy of our approach in capturing functional relationships within protein sequences. The performance achieved surpasses current baselines, highlighting the potential of our model to contribute to enzyme-catalyzed reaction classification tasks.

## 4.4 PROTEIN DESIGN

**Datasets.** We collect data from the protein structure classification database CATH. In the CATH v4.2 40% non-redundant dataset, 18024 chains are collected as the training set, 608 chains as the validation set, and 1120 chains as the test set according to the way Ingraham et al. (2019) divides the datasets. In addition, we also demonstrate the model's performance on TS50 (Li et al., 2014), a universal benchmark dataset for protein design tasks. Due to the lack of a canonical training set specifically for the TS50 test dataset, we follow the approach of (Jing et al., 2021; Qi & Zhang, 2020; Li et al., 2022) and remove 435 protein structure data similar to TS50 from the training dataset of CATH v4.2 as a new training set.

**Baselines.** We conduct a comparative analysis of our pre-trained model with various baseline approaches, encompassing specialized generative models tailored for protein design and methods focusing on protein representation learning. Structured Transformer Ingraham et al. (2019), ESM-IF Hsu et al. (2022), ProteinMPNN Dauparas et al. (2022), PiFold Gao et al. (2023) and LM-DESIGN Zheng et al. (2023b) are state-of-the-art methods for protein design, while GVP-GNN Jing et al. (2021), GBPNet Aykent & Xia (2022), DW-GCN, DW-GIN and DW-GAT Li et al. (2022) aim to construct general protein representation methods, achieving advanced performance in protein design tasks as well. With method ESM-IF utilizing CATH v4.3 for training, the remaining methods are trained using CATH v4.2. All protein representation methods employ a canonical training set for the TS50, while the training sets used by the other methods are not explicitly specified.

**Results & discussion.** According to the results shown in the Table 4, we successfully achieve the highest AAR to date on the TS50 test set, while also obtaining favorable results on the CATH v4.2 test set. In comparison to methods specifically designed for protein design, although we do not directly learn how to map structural information to sequence during the pre-training stage, the communication between the single representation and the pair representation still captures this association during fine-tuning. By comparing the TS50 test results on two different training sets, we can clearly see the significant impact of data leakage on this task. To provide a more detailed explanation of the influence of pre-training data on protein design tasks, we conduct an in-depth discussion in the appendix.

## 4.5 ANTIBODY DESIGN

**Datasets.** We collect training data from the Structural Antibody Database (SAbDab) (Dunbar et al., 2014), which contains structural data of antibody-antigen protein complexes. For the antibody sequence-structure co-design task, we partition the data according to the RefineGNN (Jin et al., 2022b) and perform sequence design and structure prediction separately for the three CDR regions of the heavy chain. For antigen-specific antibody design, we filter out protein structure data that does not contain antibody light chains or antigens. To evaluate our approach, we conduct tests on a curated benchmark dataset (Adolf-Bryfogle et al., 2018) comprising diverse CDR lengths and clusters. To prevent data leakage, any CDR sequence in the training set with over 70% identity to a CDR sequence in the test set is removed. Following preprocessing, we divide the training and validation sets based on the HSRN (Jin et al., 2022a) approach.

Table 4: Results of different Protein Design methods. [†] denotes results taken from Gao et al. (2023), [♮] denotes results taken from Li et al. (2022), and [‡] represents the results as reported in their respective papers. **Bold** and underline indicate the top two results, respectively.

| Method | CATH | | TS50 | |
|---|---|---|---|---|
| | Perplexity ↓ | Recovery % ↑ | Perplexity ↓ | Recovery % ↑ |
| Structured Transformer (Ingraham et al., 2019)† | 6.63 | 35.82 | 5.60 | 42.20 |
| ESM-IF (Hsu et al., 2022)† | 6.44 | 38.3 | - | - |
| ProteinMPNN (Dauparas et al., 2022)♮ | 4.61 | 45.96 | 3.93 | 54.43 |
| PiFold (Gao et al., 2023)† | 4.55 | 51.66 | 3.86 | 58.72 |
| LM-DESIGN(PiFold) (Zheng et al., 2023b)‡ | 4.52 | **55.65** | **3.50** | 57.89 |
| GVP-GNN (Jing et al., 2021)♮ | 5.29 | 40.2 | - | 44.9 |
| GBPNet (Aykent & Xia, 2022)‡ | 5.03 | 42.70 | - | - |
| DW-GCN (Li et al., 2022)♮ | 3.94 | 47.5 | - | 53.8 |
| DW-GIN (Li et al., 2022)♮ | **3.85** | 47.8 | - | 52.7 |
| DW-GAT (Li et al., 2022)♮ | 4.13 | 46.7 | - | 54.5 |
| ProteiNexus(canonical) | - | - | 4.78 | 59.81 |
| ProteiNexus | 5.27 | 53.45 | 4.07 | **62.15** |

**Baselines.** In the experiments involving co-design of antibody sequence and structure, we initiate our investigation by considering a sequence-based LSTM model (Saka et al., 2021; Akbar et al., 2022). This approach primarily focuses on modeling sequence information. Subsequently, we introduce RefineGNN (Jin et al., 2022b), which incorporates three-dimensional structural information and employs an iterative optimization strategy for autoregressive co-design of antibody sequence and structure. AbBERT-HMPN (Gao et al., 2022) capitalizes on an antibody pre-trained language model, enabling one-shot generation of antibody sequences. Additionally, we employ a multi-round 3D equivariant model MEAN (Kong et al., 2023a).

**Results & discussion.** In the context of framework region conditioned design, our approach demonstrates a clear superiority over existing baselines, showcasing our model's ability to extract information from contextual cues. With the incorporation of antigen and light chain information, we successfully achieve precise generation of antibody sequences and structures for both CDR-H3 and all six CDRs (as shown in Table 8 and 9 in the appendix). This achievement highlights our model's significant advancement in comprehensively considering information from various levels.

Table 5: Results of Antibody Design: Sequence-Structure Co-design. The best and the runner-up results are highlighted in **bolded** and underlined, respectively.

| Method | CDR-H1 | | CDR-H2 | | CDR-H3 | |
|---|---|---|---|---|---|---|
| | AAR % ↑ | RMSD ↓ | AAR % ↑ | RMSD ↓ | AAR % ↑ | RMSD ↓ |
| LSTM (Saka et al., 2021; Akbar et al., 2022) | 28.02 | - | 24.39 | - | 18.92 | - |
| AR-GNN (Jin et al., 2020) | 41.88 | 2.87 | 41.18 | 2.34 | 18.93 | 3.19 |
| RefineGNN (Jin et al., 2022b) | 30.07 | 0.97 | 27.70 | 0.73 | 27.60 | **2.12** |
| AbBERT-HMPN (Gao et al., 2022) | 55.56 | **0.91** | 51.46 | **0.67** | 31.08 | 2.38 |
| MEAN (Kong et al., 2023a) | 62.78 | 0.94 | 52.04 | 0.89 | 39.87 | 2.20 |
| ProteiNexus | **64.18** | 1.57 | **58.05** | 1.62 | **41.01** | 3.06 |

## 5 CONCLUSION

In this work, we introduce an efficient pre-training model named ProteiNexus, capable of parallelly capturing both protein sequence and structural information. We integrate a hybrid structural encoding and self-supervised prediction strategy to obtain meaningful representations, and successfully apply them to various downstream tasks. Experimental results confirm the outstanding performance of our approach across a range of tasks, particularly in the understanding of protein complexes. In the future, we intend to extend ProteiNexus to a broader range of applications, addressing more practical problems.

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

# A MORE RELATED WORK

## A.1 PROTEIN STRUCTURE MATTERS

Protein structure is essential to tackle most downstream tasks. This is underscored by the complexity of the protein folding problem in the field of biology. This signifies that even when two proteins share similar amino acid sequences, they can fold into entirely distinct three-dimensional structures. This discrepancy becomes particularly apparent during post-translational modifications following protein translation, such as glycosylation, phosphorylation, methylation, acetylation, which profoundly alter the protein's structure and function. Anomalies in these modifications can even lead to serious diseases like leukemia, pancreatic dysfunction, and Alzheimer's disease (Mehboob & Lang, 2021). In the context of Alzheimer's disease, for instance, a portion of beta-amyloid protein may form toxic plaques due to misfolding, exerting detrimental effects on neural cells (Hamley, 2012; Wang et al., 2022). Furthermore, G-protein-coupled receptors (GPCRs) in proteins undergo conformational changes in their extracellular regions upon binding with excitatory signaling molecules like odors, hormones, neurotransmitters, and chemokines (Che et al., 2020; He et al., 2021b). Figure 3 presents a specific example illustrating the conformational changes that occur in the $G\alpha$ subunit (comprising two subdomains, the Ras domain and the AHD domain) during receptor-mediated G protein nucleotide exchange. This further accentuates the critical role of protein structure in regulating biological functions.

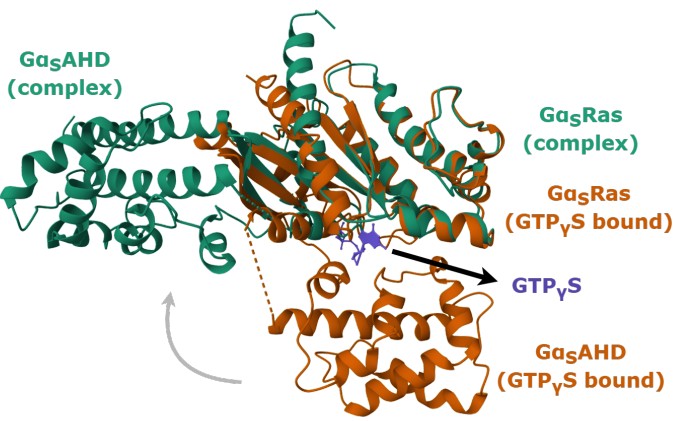

Figure 3: **Interaction-mediated conformational changes.** The figure depicts structural changes between receptor-bound and nucleotide-free $G\alpha_s$ (Turquoise, PDBID 3SN6) and $G\alpha_s$(Burnt Sienna, PDBID 1AZT) bound to $GTP_\gamma S$ (Indigo). Research has revealed that the receptor for $G_s$ induces a movement of the $\alpha$-helical domain ($G\alpha_s AHD$) of $G\alpha_s$, causing it to shift outward relative to its position in the $GTP_\gamma S$-bound state, thereby triggering conformational changes (the receptor of $G_s$ is not shown in the figure). This example is derived from Hilger et al. (2018).

## A.2 PROTEIN STRUCTURE REPRESENTATION LEARNING

Given the critical role of protein structure in determining function, structure-based representation methods emerge as a superior solution. In the past, these methods often rely on manually designed feature extraction techniques, such as using Voronoi tessellation to describe protein contact areas (Olechnovič & Venclovas, 2017) or employing 3D Zernike descriptors to characterize protein surface properties (Sael et al., 2008; Venkatraman et al., 2009; Daberdaku & Ferrari, 2018). Although these methods are effective to some extent, they struggle to capture complex protein structural information. With the advancement of deep learning, a new generation of methods continuously emerges. In the early stages, 3D Convolutional Neural Networks (3D CNNs) are employed to voxelate protein structures (Amidi et al., 2018; Derevyanko et al., 2018). Subsequently, Graph Neural Networks (GNNs) gain prominence by abstracting protein structures into graphs. Some methods even integrate multiple general GNN frameworks to introduce geometric information Jing

et al. (2021) or maintain SO(3)-equivariance properties (Li et al., 2022), aiming for a more precise representation of protein structures. Furthermore, the representation of local protein structures also garners significant attention. For instance, some methods concentrate on extracting information from the protein surface, as seen in MaSIF (Gainza et al., 2020) and dMaSIF (Sverrisson et al., 2021). This is crucial for identifying potential protein-protein interaction interfaces. Uni-Mol (Zhou et al., 2023), on the other hand, focuses on learning universal representations, with particular emphasis on pseudo protein pockets that could form interfaces.

## B  MODEL DETAILS

### B.1  ENCODER

**Spatial Position Encoding.**     In this section, we delve into the further details of Spatial Position Encoding. Here, $\vec{x}_{i,1}$, $\vec{x}_{i,2}$, and $\vec{x}_{i,3}$ represent the coordinates of N, C$\alpha$, and C atoms in the $i$-th residue, while $\vec{x}_{j,2}$ denotes the C$\alpha$ atom in the $j$-th residue. As illustrated in Figure 4, we establish a local Cartesian coordinate system $\mathcal{O}_i$ based on $\vec{x}_{i,1}$, $\vec{x}_{i,2}$, and $\vec{x}_{i,3}$. $\vec{d}_{ij}$ corresponds to the coordinates of $\vec{x}_{j,2}$ in $\mathcal{O}_i$, encapsulating the relative positional relationship between two residues. Algorithm 1 elucidates the specific operations of SPE, with $\lfloor \ \rfloor$ denoting the binning process, which categorizes $\vec{r}_{ij}$ into $v_{bins}$. Considering that intermolecular forces significantly decrease as distances exceed a certain threshold, we set a cutoff for this purpose.

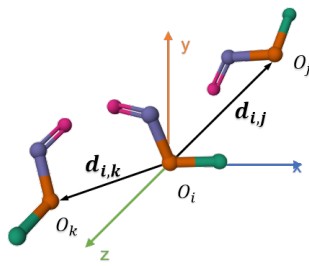

Figure 4: The sketch map of Spatial Position Encoding(SPE), with N, C$\alpha$, C, and O colored in teal, burnt orange, periwinkle, and magenta, respectively.

---

**Algorithm 1** Spatial Position Encoding (SPE)

---

**Require:** $v_{bins} = [0, ..., 128], \vec{x}_{i,1}, \vec{x}_{i,2}, \vec{x}_{i,3}, \vec{x}_{j,2} \in \boldsymbol{R}^3$

1: $\vec{v}_{i,1} = \vec{x}_{i,1} - \vec{x}_{i,2}; \vec{v}_{i,2} = \vec{x}_{i,3} - \vec{x}_{i,2}$

2: $\vec{e}_{i,1}, \vec{e}_{i,2} = \text{Gram-Schmidt}(\vec{v}_{i,1}, \vec{v}_{i,2})$          ▷ Compute an orthogonal basis

3: $\vec{e}_{i,3} = \vec{e}_{i,1} \times \vec{e}_{i,2}$

4: $\mathcal{O}_i = \text{concat}(\vec{e}_{i,1}, \vec{e}_{i,2}, \vec{e}_{i,3})$          ▷ Local frame constructed by the $i$-th residue

5: $\vec{d}_{ij} = \vec{x}_{i,2} - \vec{x}_{j,2}$

6: $\vec{r}_{ij} = \text{concat}(\|\vec{d}_{ij}\|, \vec{d}_{ij} \circ \mathcal{O}_i)$          ▷ $\circ$ represent the projection of $\vec{d}_{ij}$ in the local frame $\mathcal{O}_i$

7: $p_{ij} = \text{Linear}(\text{one\_hot}(\lfloor \vec{r}_{ij} \rfloor, v_{bins})$

8: **return** $p_{ij}$

---

**Distance Encoding.**     This method involves converting the coordinates of backbone atoms into a distance matrix and discretizes continuous distance values into distinct bins using fixed distance thresholds. As atom distances increase significantly, the intermolecular forces between them diminish. In such instances, all distances exceeding a certain threshold are categorized within the maximum distance bin. To facilitate hierarchical learning at various precision levels for distance representation, we discretize distances into different-sized distance bins, where $|\mathcal{V}| = \{16, 64, ..., 16384\}$, and $\mathcal{V} = \{0, 1, ..., |\mathcal{V}| - 1\}$ represents the distance vocabulary. Subsequently, linear layers are employed to embed the distance intervals at each level, followed by an aggregation step to obtain the

initial distance representation. This hierarchical learning approach allows for the extraction of more nuanced and fine-grained distance representations, enhancing the model's ability to capture subtle structural features and relationships within residues.

Our atom-level distance representations obtained through distance encoding encompass the relative orientations of backbone atoms. Similarly, the residue-level pair representations acquired via relative spatial encoding consider both residue orientation and inter-residue spatial relationships. To amalgamate these representations across two levels, we devise a purposeful projection that accurately aligns atom-level representations with their corresponding residue-level counterparts. This alignment mechanism facilitates the transmission and matching of information, thereby ensuring comprehensive structural modeling across multiple hierarchical levels.

## C FINE-TUNING ON DOWNSTREAM TASKS

### C.1 MODEL QUALITY ASSESSMENT

**Model architecture** We utilize the pre-trained model as the backbone and employ a layer consisting of a two-layer MLP as the predictor. Our objective is to predict the quality of both global and local structures. Initially, we conduct column-wise and row-wise aggregation on pair representations, then concatenate these aggregated representations with single representations and use a MLP along with the sigmoid function to map them into the (0,1) range, signifying scores for each residue. For assessing global structural quality, we follow the same procedure, ultimately averaging the scores at the residue level to derive the global score.

**Datasets.** We constructed a training dataset comprising 39,922 decoys (corresponding to 7,992 native structures). While generating a significant number of decoys to expand the dataset was feasible, we observed that the diversity inherent in native structures proved more effective during training. As depicted in Table 6, we selected two datasets of equal size, where one encompassed decoys corresponding to 7,992 native structures, and the other contained decoys corresponding to 270 native structures (with more decoys per native structure). Although both training datasets are of comparable scale, models trained with the diversity of native structures exhibit superior generalization capabilities. This underscores the critical importance of accurately representing native structures in the learning process. Furthermore, even scaling up the dataset, including training with and without pretraining using the entire DeepAccNet dataset, doesn't yield substantial improvements. This further underscores the robust representation capabilities of our model, which only requires simple fine-tuning on a small dataset to achieve optimal performance.

Table 6: Comparison of Model Quality Assessment on different training sets.

| DATASETS | CASP14 | | | | | | | | CASP15 | | | | | | | |
| | GDT-TS | | | | LDDT | | | | GDT-TS | | | | LDDT | | | |
| | RMSE ↓ | $\mathcal{P}$ | $\mathcal{S}$ | $\mathcal{K}$ | RMSE ↓ | $\mathcal{P}$ | $\mathcal{S}$ | $\mathcal{K}$ | RMSE ↓ | $\mathcal{P}$ | $\mathcal{S}$ | $\mathcal{K}$ | RMSE ↓ | $\mathcal{P}$ | $\mathcal{S}$ | $\mathcal{K}$ |
|---|---|---|---|---|---|---|---|---|---|---|---|---|---|---|---|---|
| Diversity test | 0.17 | 0.70 | 0.70 | 0.52 | 0.12 | 0.75 | 0.77 | 0.58 | 0.25 | 0.74 | 0.74 | 0.55 | 0.25 | 0.60 | 0.58 | 0.41 |
| DeepAccNet w/o pretraining | 0.16 | 0.74 | 0.75 | 0.56 | 0.10 | 0.77 | 0.79 | 0.60 | 0.20 | 0.74 | 0.67 | 0.48 | 0.20 | 0.56 | 0.56 | 0.40 |
| DeepAccNet | 0.17 | 0.72 | 0.72 | 0.53 | 0.11 | 0.75 | 0.76 | 0.57 | 0.17 | 0.78 | 0.75 | 0.55 | 0.13 | 0.72 | 0.68 | 0.49 |
| ProteiNexus | **0.14** | **0.79** | **0.78** | **0.59** | **0.10** | **0.83** | **0.82** | **0.63** | **0.15** | **0.84** | **0.83** | **0.63** | **0.13** | **0.79** | **0.72** | **0.53** |

To ensure impartial model performance evaluation, we selected targets from the most recent two rounds of The Critical Assessment of protein Structure Prediction(CASP) for our test set. Our evaluation focuses on monomer structures, although our approach can easily be extended to assess the quality of multimer structures. To obtain GDT-TS and LDDT scores for predicted structures, we clipped the targets based on experimentally resolved native structures, discarding predicted structures with sequence lengths inconsistent with the native structures. Due to the excessive length of the sequence for target T1169 in CASP15, baseline methods encountered inference difficulties, prompting us to exclude it from the test set. The remaining target IDs included in the test set are summarized in Table 7. The original predicted structures for each target can be accessed through publicly available links: `https://predictioncenter.org/download_area/CASP14/predictions` and `https://predictioncenter.org/download_area/CASP15/predictions`. We calculated GDT-TS and LDDT scores using publicly available tools, which can be downloaded and installed

from `https://zhanggroup.org/TM-score/` and `conda install -c bioconda lddt`, respectively.

Table 7: Target IDs in the Model Quality Assessment Test Set.

| DATASETS | Target ID List | Total |
|---|---|---|
| CASP14 | T1024, T1025, T1026, T1027, T1028, T1029, T1030, T1031, T1033, T1035, T1036s1, T1037, T1039, T1040, T1041, T1042, T1043, T1045s1, T1045s2, T1046s1, T1046s2, T1047s1, T1047s2, T1049, T1051, T1053, T1055, T1056, T1057, T1058, T1059, T1060s2, T1060s3, T1064, T1065s1, T1065s2, T1072s1, T1072s2, T1074, T1076, T1082, T1089, T1090, T1091, T1092, T1093, T1094, T1095, T1096, T1099 | 50 |
| CASP15 | T1104, T1120, T1133, T1159, T1169, T1119, T1121, T1123, T1124, T1152, T1170, T1187, T1106s1, T1106s2, T1114s1, T1114s2, T1114s3, T1129s2, T1134s1, T1134s2, T1137s1, T1137s2, T1137s3, T1137s4, T1137s5, T1137s6, T1137s7, T1137s8, T1137s9 | 29 |

**Evaluation metrics.** When the native structure is known, there are multiple evaluation methods that can measure the degree of similarity between the predicted structure and the native structure, that is, the quality of the predicted structure. We predicted the GDT-TS score to evaluate the overall quality of the model and the LDDT score to evaluate the quality of each residue in the absence of the native structure. Root Mean Square Error and three statistical correlation coefficients, Pearsons correlation $r$, Spearmans $\rho$, and Kendalls $\tau$ were used to evaluate the accuracy of the predicted score.

## C.2 BINDING AFFINITY

**Model architecture** The change in binding affinity is calculated by the formula $\Delta\Delta G = \Delta G_{\text{wild\_type}} - \Delta G_{\text{mutant}}$. Given the assumption that the structure of wild-type and mutant structures does not undergo significant changes, we exclusively consider the single representations to compute the change in binding affinity values, as shown below:

$$\Delta\Delta g = \text{avg}(\boldsymbol{I}_\psi(\text{Linear}(\text{MLP}(f_{wm}^i) - \text{MLP}(f_{mw}^i))))) \tag{3}$$

where $\boldsymbol{I}$ is the indicator function that equals 1 when $i \in \psi$, the set $\psi$ represents the indices of mutant residues. $f_{wm}^i = \text{concat}(s_{w,i}^L, s_{m,i}^L)$, $f_{mw}^i = \text{concat}(s_{m,i}^L, s_{w,i}^L)$, where $s_{m,i}^L$ and $s_{w,i}^L$ respectively denote the single representations of the $i$-th wild-type and mutant residues in the final layer output.

**Evaluation metrics.** We utilize Pearson correlation coefficient (Rp) and **R**oot **M**ean **S**quare **E**rror (RMSE) as evaluation metrics to quantify the disparity between predicted binding affinity values and ground-truth. The Pearson correlation coefficient assesses the degree of linear relationship between prediction and ground-truth, with a value closer to 1 indicating a stronger linear relationship. On the other hand, RMSE measures the average magnitude of deviations between predicted and ground-truth, with a smaller value indicating higher prediction accuracy.

## C.3 FOLD AND ENZYME-CATALYZED REACTION CLASSIFICATION

**Model architecture** We employ a straightforward linear layer as the classifier for our classification task. we obtain the representation $\boldsymbol{h}_i^L$ by processing the final single representation $\boldsymbol{s}_i^L$ through a fully connected layer and an activation function, followed by normalization. The probability for each individual category is computed using $\text{softmax}(\text{avg}(\{\boldsymbol{h}_i^L\}_{i=1}^n \boldsymbol{W}_c + \boldsymbol{b}_c))$, where $\{\boldsymbol{h}_i^L\}$ signifies the final single representation of the $i$-th residues, $c$ represents the number of classes $\boldsymbol{W}_c$ denotes the learnable parameter matrix, and $\boldsymbol{b}_c$ stands for the bias term. In the fold classification task, $c = 1195$, indicating 1195 identified folds. In the Enzyme-Catalyzed Reaction Classification task, $c = 384$, representing 384 different Enzyme Commission numbers.

**Evaluation metrics.**   We assess the model's classification performance using classification accuracy, which indicates the proportion of all predictions that are successfully classified into the correct category.

## C.4   PROTEIN DESIGN

**Model architecture**   We utilize [MASK] to denote the residue types at each position. Leveraging the structural encoder, we transmit the backbone structural information to the single representation. Subsequently, we apply a task layer, similar to the one used in classification tasks, to predict the residue types, with $c$ indicating the size of the residue type dictionary.

**Evaluation metrics.**   For evaluating protein sequence generation tasks, we employ perplexity and **A**mino **A**cid **R**ecovery(AAR) as evaluation metrics. Perplexity quantifies the model's uncertainty during sequence generation, where lower perplexity values signify closer alignment between the model's predictions and the native sequence. Amino Acid Recovery measures the proportion of amino acids in the generated sequence that match the target sequence. A higher Amino Acid Recovery indicates a higher similarity between the model's generated sequence and the target sequence, which reflects better performance of the model.

## C.5   ANTIBODY DESIGN

**Model architecture**   Apart from generating sequences for the Complementarity-Determining Regions(CDRs), we introduce the structure module to predict the structure of regions with unknown sequences. We undertake work in two primary areas: sequence-structure co-design and antigen-specific antibody design.

- **Sequence-Structure Co-design Task.**   Our primary focus is on the antibody's heavy chain. As an example, for the design of CDR-H3, we renumber the antibody heavy chain using the IMGT to precisely locate CDR-H3 within the sequence. We mask the residues belonging to CDR-H3 and assign coordinates to these residues by taking the average of the C$\alpha$ coordinates of the two nearest residues outside this region. This process results in initial pair representations with noise. Subsequently, we employ the pre-trained model to predict the residue types of CDR-H3. These predictions, along with the updated pair representations, are fed into the structure module. We use a combination of cross-entropy loss, smooth l1 loss and frame-aligned point error as the loss functions for sequence generation and structure generation, with equal weighting 1:1:1.

- **Antigen-Specific Antibody Design Task.**   We initially assess our capability to generate CDR-H3 on the well-established Benchmark RAbD dataset. In this process, we introduce both the antigen's sequence and structural information while retaining the sequence and structural information of the antibody heavy chain framework region. Notably, we assume that the relative positions of the antigen and the antibody heavy chain are unknown, implying a lack of inter-chain information. To construct the initial representations of the antibody-antigen complex, we separately obtain single and pair representations for the antigen and the antibody heavy chain using the pre-trained model. The single representations are concatenated to obtain the complex's single representation. For the complex's pair representation, the positions along the diagonal (representing intra-chain information) are replaced with the pair representations of the antigen and the antibody heavy chain. However, the positions along the anti-diagonal (representing inter-chain information) remain empty. Subsequently, we introduce a model identical to the pre-training model to update the complex's representation, thereby completing the inter-chain information. The updated complex representation is then input into the structure module to predict the unknown structure of CDR-H3.

In fact, our approach can be straightforwardly extended to simultaneously predict all six CDRs regions of antibody-antigen complexes, as demonstrated in Table 9, once the sequence and structural information of the light chain variable region is introduced.

**Baselines.**   Expanding our scope to encompass the design of antigen-specific binding antibodies, we introduce an additional set of methodologies. Among these, we incorporate the physics-based traditional approach RAbD  (Adolf-Bryfogle et al., 2018). Furthermore, we integrate the hierarchical

Table 8: Results of Antibody Design: Antigen Specific Design. The best and the runner-up results are highlighted in **bolded** and underlined respectively.

| Model | AAR % ↑ | RMSD ↓ |
|---|---|---|
| RAbD  (Adolf-Bryfogle et al., 2018) | 28.6 | - |
| LSTM  (Saka et al., 2021; Akbar et al., 2022) | 22.36 | - |
| CondRefineGNN  (Jin et al., 2022b) | 33.2 | - |
| HSRN  (Jin et al., 2022a) | 34.1 | - |
| MEAN  (Kong et al., 2023a) | 36.77 | **1.81** |
| dyMEAN  (Kong et al., 2023b) | **43.65** | - |
| ProteiNexus | 42.33 | 2.25 |

Table 9: One-shot generates results for the antibody design of six CDRs simultaneously.

| Model | CDR-L1 | CDR-L2 | CDR-L3 | CDR-H1 | CDR-H2 | CDR-H3 |
|---|---|---|---|---|---|---|
| dyMEAN | 73.55 | 83.10 | 52.12 | 75.72 | **68.48** | 37.51 |
| ProteiNexus | **78.19** | **84.86** | **72.21** | **77.33** | 68.34 | **39.58** |

model HSRN  (Jin et al., 2022a), tailor-made for antibody-antigen interface design. To enhance the design of antibody heavy chains, MEAN, the end-to-end 3D equivariant model dyMEAN, and diffusion-based model DiffAB not only consider antigen but also incorporate antibody light chain information into the known conditions.

**Evaluation metrics.** We employ **A**mino **A**cid **R**ecovery (AAR) and **R**oot **M**ean **S**quare **D**eviation (RMSD) as key evaluation metrics to assess the quality of generated complementarity-determining regions (CDRs). The AAR reflects the similarity between the generated CDR sequence and the target sequence, quantifying the proportion of successfully recovered target amino acids within the generated CDR, thereby capturing sequence-level quality. On the other hand, RMSD focus on the spatial configuration of CDR structures. RMSD measures the average atomic coordinate deviation between the generated CDR structure and the target structure.

# D    EXPERIMENTS DETAIL & REPRODUCE

## D.1    DATASETS

Table 10 showcases the dataset statistics for both pre-training and downstream tasks, with data splitting principles primarily drawn from well-established benchmarks in the field. Further details are provided below.

**Pre-training.** Our pre-training dataset is sourced from the Protein Data Bank (PDB) database, encompassing all protein structure data released up until May 1st, 2023. We conduct rigorous data filtering and cleaning, excluding elements such as RNA, DNA, small molecules, water molecules, and heterogeneous residues from the PDB files. Additionally, we complete residues with missing backbone atom coordinates. Subsequently, we randomly split the data into training and validation sets in a 9:1 ratio. Although the objective of pre-training slightly differs from that of protein design, we take extra measures to prevent potential data leakage. Specifically, we perform additional data processing by creating a pre-training validation dataset composed of the CATH v4.2 test set and the TS50 test set, while the remaining data is included in the training set. This supplementary processing is intended for ablation experiments to confirm the absence of data leakage.

## D.2    PRE-TRAINING IMPLEMENTATION DETAILS

For the two self-supervised tasks corresponding to pre-training, namely 'masked residue type prediction' and 'pair representation denoising', we employ two distinct loss functions, specifically cross-entropy loss and Smooth L1 loss. To facilitate effective model training, we combine these two loss

Table 10: Dataset statistics for pre-train and downstream tasks.

| DATASETS | # TRAIN | # VALID | # TEST | TASK |
|---|---|---|---|---|
| Pre-training | 176401 | 19600 | - | - |
| Pre-training - *Data Leakage* | 175395 | 19476 | - | - |
| Model Quality Assessment - *CASP14* | 35,176 | 3,881 | 24,313 | Regression |
| Model Quality Assessment - *CASP15* | 35,176 | 3,881 | 13,260 | Regression |
| Binding Affinity - *S1131* | 907 | 111 | 111 | Regression |
| Binding Affinity - *S4169* | 3,341 | 414 | 414 | Regression |
| Binding Affinity - *S8338* | 6,680 | 829 | 829 | Regression |
| Binding Affinity - *M1101* | 824 | 102 | 102 | Regression |
| Binding Affinity - *M1707* | 1,150 | 143 | 143 | Regression |
| Fold Classification - *Fold* | 12,312 | 736 | 718 | Classification |
| Fold Classification - *Superfamily* | 12,312 | 736 | 1,254 | Classification |
| Fold Classification - *Famliy* | 12,312 | 736 | 1,272 | Classification |
| Enzyme-Catalyzed Reaction Classification | 29,215 | 2,562 | 5,651 | Classification |
| Protein Design - *CATH v4.2* | 18,024 | 608 | 1,120 | Generation |
| Protein Design - *TS50* | 18,024 | 608 | 50 | Generation |
| Protein Design - *TS50(canonical)* | 17,669 | 577 | 50 | Generation |
| Antibody Design - *CDR-H1* | 4,050 | 359 | 326 | Generation |
| Antibody Design - *CDR-H2* | 3,876 | 483 | 376 | Generation |
| Antibody Design - *CDR-H3* | 3,896 | 403 | 437 | Generation |
| Antibody Design - *RAbD* | 2,237 | 155 | 56 | Generation |

functions with equal weights of 1:1, constituting the overall loss function during the pre-training phase. All models are trained on 8 NVIDIA A100 40GB GPUs. Additionally, further hyperparameter configurations related to pre-training can be found in Table 11.

Table 11: Hyperparameters setup during pre-training

| Hyperparameters | Base Size |
|---|---|
| Layers | 15 |
| Hidden size | 512 |
| FFN hidden size | 2048 |
| Attention heads | 4 |
| Attention head size | 128 |
| Training epochs | 500 |
| Batch size | 32 |
| Adam $\epsilon$ | 1e-12 |
| Adam $\beta$ | (0.9, 0.82) |
| Peak learning rate | 1e-4 |
| Learning rate schedule | polynomial |
| Warmup steps | 5000 |
| Gradient clip norm | 1.0 |
| Dropout | 0.1 |
| Weight decay | 1e-4 |
| Activation function | GELU |
| Sequence crop size | 256 |
| Spatial crop ratio | 0.5 |
| Mask ratio | (0.15, 0.5, 1.0) |
| Mask ratio probability | (0.6, 0.2, 0.2) |
| Noise type | $\mathcal{N}(0, 0.1), \mathcal{N}(0, 1)$ |
| Noise probability | (0.2, 0.8) |
| Vocabulary size (residue types) | 24 |

## D.3 Downstream task Implementation Details

We previously mention an overview of the task layer and datasets used during the fine-tuning of downstream tasks. Due to space constraints, we provide a more detailed exposition in this section. Throughout the fine-tuning process for various downstream tasks, we train our models with a dropout rate of 0.2 and a warm-up ratio of 0.06. All training is conducted on 8 NVIDIA V100 32GB GPUs. Additionally, we summarize the differences among settings for different downstream tasks, as shown in Table 12. Further details are presented below.

Table 12: Hyperparameters setup during fine-tuning.

| Task | Epoch | Batch Size | Learning Rate | Loss |
|------|-------|-----------|---------------|------|
| Model Quality Assessment | 1 | 64 | 5e-4 | MSE |
| Binding Affinity | 100 | 16 | 3e-4 | MSE |
| Fold Classification | 100 | 32 | 5e-4 | Cross entropy |
| Enzyme-Catalyzed Reaction Classification | 100 | 32 | 5e-4 | Cross entropy |
| Protein Design | 20 | 64 | 1e-4 | Cross entropy |
| Antibody Design | 40 | 8 | 3e-4 | Cross entropy & Smooth L1 loss & FAPE |

# E Ablation Study

We conduct comprehensive ablation experiments to verify the effectiveness of each component of the pre-trained model. Our primary focus lies in validating the results of these ablation experiments through classification tasks and protein design. Initially, we scrutinize the most critical encoder ablation to assess its effectiveness in structural representation. We delve into the effectiveness of pre-training, exploring the influence of pre-training data and strategies. Lastly, we analyze the potential existence of data leakage.

Table 13: The results of the ablation study. The first segment pertains to encoder ablation, while the second segment corresponds to pre-training ablation. ✓ signifies that the respective component is enabled, while ✗ indicates its deactivation. Metrics for the classification task are represented by mean accuracy, whereas for protein design, validation is solely conduct on the CATH v4.2 test set with metrics measured as AAR.

| | Modifications | | | | | | Results | | | | |
|---|---|---|---|---|---|---|---|---|---|---|---|
| | Encoder | | | Data Level | Noise Type | Pre-training | Fold | | | EC | CATH |
| | SPE | Distance | RPE | | | | Fold | Sup | Family | | |
| Experiment 1 | ✗ | ✓ | ✓ | backbone atoms | mix | ✓ | 46.8 | 80.5 | 98.0 | 86.4 | 52.2 |
| Experiment 2 | ✓ | ✗ | ✓ | backbone atoms | mix | ✓ | **51.9** | **81.7** | 98.0 | 86.1 | 40.8 |
| Experiment 3 | ✓ | ✓ | ✗ | backbone atoms | mix | ✓ | 43.9 | 77.8 | 97.8 | **88.9** | 49.0 |
| Experiment 4 | ✗ | ✗ | ✓ | backbone atoms | mix | ✓ | 15.5 | 25.5 | 86.6 | 68.8 | - |
| Experiment 5 | ✗ | ✓ | ✓ | Cα | mix | ✓ | 48.5 | 79.7 | **98.1** | 88.2 | 43.7 |
| Experiment 6 | ✓ | ✓ | ✓ | backbone atoms | mix | ✗ | 17.1 | 25.7 | 85.1 | 46.9 | 32.1 |
| Experiment 7 | ✓ | ✓ | ✓ | backbone atoms | single | ✓ | 38.7 | 73.6 | 97.9 | 87.0 | 40.5 |
| ProteiNexus | ✓ | ✓ | ✓ | backbone atoms | mix | ✓ | 47.6 | 79.7 | 98.0 | 88.4 | **53.5** |

## E.1 ENCODER ABLATION

While we consider spatial position encoding (SPE), distance encoding, and relative position encoding (RPE) for protein structure as an integrated whole, with each component playing a crucial role, we conduct experiments 1-4 to assess their individual contributions, as presented in Table 13. Initially, we disable each component separately for validation. Subsequently, we simultaneously deactivate SPE and distance encoding, essentially depriving the model of its structural awareness module. Therefore, we opt not to validate this configuration for protein design tasks. Experimental results demonstrate that, despite the relatively low reliance on protein structure information in classification tasks, the removal of a robust structural representation encoder still significantly impacts the results. This impact becomes more pronounced in tasks such as protein design that rely entirely on structural representations.

### E.2   PRE-TRAINING ABLATION

**Backbone atoms v.s C$\alpha$.**   We aim to validate the necessity of incorporating coordinates for backbone atoms. In experiment 5, we employ C$\alpha$ atom coordinates to represent the positions of residues and conduct the corresponding pre-training. It's important to note that, given our sole reliance on C$\alpha$ atom coordinates, we cannot establish a local frame for each residue, so Spatial Position Encoding is no longer utilized in these experiments to enhance structural information. The experimental results underscore that the inclusion of backbone atom coordinates enriches the representation of protein structures, providing crucial support for residue orientation and spatial positional information, thereby enhancing the model's performance in downstream tasks.

**w pre-training v.s w/o pre-training.**   In Experiment 6, we conduct an assessment of the effects of pre-training models on large-scale datasets. Acquiring labeled data can be a costly endeavor in many tasks, and often there isn't a sufficient amount of data available to support effective model training. This limitation can hinder the model's ability to generalize effectively. However, by pre-training on extensive datasets, the model can learn more accurate representations, leading to improvements in its performance. Comparative results between experiments with and without pre-training demonstrate that this pre-training approach enables the model to better adapt to various tasks, thereby enhancing its generalization capability and practicality.

**Pre-training strategy.**   Furthermore, we delve into different pre-training strategies. We explore two strategies: a single masking strategy and a mix noise strategy. Specifically, we randomly mask 15% of sequence residues and introduce noise uniformly distributed within the range of (-1,1) to atom coordinates. Results demonstrate that the mixed training strategy was more conducive to fostering interactions between one-dimensional sequence and three-dimensional structural information, as well as enhancing the model's capability to infer correct structural representations from contextual information. In comparison to the single strategy, it exhibits superior performance.

### E.3   DATA LEAKAGE

Pre-trained self-supervised tasks and downstream tasks in protein design are analogous. In comparison to tasks with additional data labels, concerns arise regarding potential data leakage. To address this concern, we conduct a series of ablation studies to elucidate the situation. It's worth emphasizing that due to the fact that the binding of antibody-antigen complexes typically relies on electrostatic interactions and has not undergone extended evolutionary processes, the impact of our pre-trained model on antibody design tasks is relatively modest when trained on generic protein data. In other words, even if we start training from scratch, we can achieve performance on par with what we describe in the main text.

To investigate the impact of data leakage on protein design, we reprocess the pre-trained data, following the methodology outlined in section D.1. Our experimental results are presented in Table 14. Notably, the removal of a small fraction of the pre-trained data has a substantial impact on the results, suggesting the potential presence of data leakage in protein design. However, it is important to emphasize that during the fine-tuning stage, we incorporate a new prediction layer rather than utilizing the layer responsible for predicting residue types from the pre-training, despite the fact that these two layers share the same architecture. This implies that during the fine-tuning phase, we reacquire the capability to map a single representation to residue types. Furthermore, judging by the extent of the impact of data leakage on the CATH test set and TS50 test set, the deterioration in results is more likely attributable to the removal of data that influenced the distribution of pre-training data. We will conduct a comprehensive range of experiments to mitigate the effects arising from data distribution imbalances.

Table 14: The results of ablation study on data leakage in protein design tasks.

| Setting | CATH | | TS50 | |
|---|---|---|---|---|
| | Perplexity ↓ | Recovery % ↑ | Perplexity ↓ | Recovery % ↑ |
| Pre-training - *Data Leakage* | 7.81 | 43.49 | 4.99 | 60.30 |
| ProteiNexus | 5.27 | 53.45 | 4.07 | 62.15 |

