# OpenReview forum: "ProteiNexus: Illuminating Protein Pathways through Structural Pre-training"
_ICLR.cc/2024/Conference — Submitted to ICLR 2024_

### Official Review · Reviewer_oPJy · 2023-10-23

**Soundness:** 1 poor
**Presentation:** 2 fair
**Contribution:** 1 poor
**Rating:** 3
**Confidence:** 5

**Summary:**

The paper introduces a general architecture for protein representation learning, utilizing a pre-training method of masked residue type prediction. Following this, the model is fine-tuned using lightweight decoders for a range of downstream tasks such as model quality assessment, binding affinity change prediction, EC and fold classification, protein design, and antibody design. It manages to achieve state-of-the-art performance in certain areas.

**Strengths:**

The paper's ambition to create a universal protein pre-training model is commendable. Pursuing this aim, the author presents a transformer architecture that seamlessly integrates sequence and structural data through a straightforward pre-training objective.

**Weaknesses:**

1. The majority of the components within the proposed method are adaptations from prior studies, which the paper fails to acknowledge. The transformer approach mirrors that of [1], and the pre-training objective resembles [2]. Consequently, the work's novelty is somewhat questionable.
2. The paper's "related work" section is not exhaustive. Notable omissions in the protein language domain include TAPE [3], ProtTrans [4], Ankh [5], and ProGen2 [6]. Additionally, recent advancements in protein structure representation, such as ProNet [7] and CDConv [8], are overlooked. A discussion on the connection between this study and previous ones is conspicuously absent.
3. The presented experimental results have serious issues like data leakage and the absence of critical baselines, undermining the paper's claims.

For details, please refer to the Question section.

**Questions:**

1. The authors motivate the use of transformers as encoders by critiquing graph-based representation learning methods for two reasons: 1) their inability to capture detailed atomic information, and (2) their disregard for long-range interactions. However, recent studies [7,9,10] demonstrate that graph-based methods can indeed be extended to the atom level. Additionally, the authors' method focuses solely on backbone-level structures, thus overlooking side-chain details. Regarding long-range interactions, the paper lacks experiments substantiating their claim. Thus, the critiques they raise against graph-based methods lack evidence.
2. It's inadvisable for the authors to describe their method as "groundbreaking" in the introduction—this is a clear exaggeration.
3. The model quality assessment, used by the authors for evaluation, has a potential data leakage risk during pre-training. This task aims to predict the GDT-TS score of certain model predictions without revealing the ground-truth structures. Yet, using the PDB data up to May 1st, 2023, means the model may have encountered target structures from CASP14 and CASP15 during pre-training. Despite different loss functions, this could pose a significant issue.
4. In the model quality assessment, the authors omit essential baselines. Notably, this task has been included in the Atom3D benchmark [11], where baselines [9,12] are essential references.
5. For binding affinity prediction, the authors neglect to explain their dataset splits—critical to avoid data leakage. Given the small test datasets, it's standard to conduct multiple cross-validations under varying random seeds. Traditional methods like FlexDDG [13] should also be considered for comparison.
6. In the EC and fold classification benchmarks, there's an absence of vital baselines, notably CDConv [8] and ESM-GearNet [14]. The authors might also explore the more challenging GO prediction tasks detailed in [15]. Even without these benchmarks, the authors' method falls behind leading approaches.
7. For protein design tasks, there are serious data leakage problems due to the presence of test data in pre-training dataset. As discussed in App. E.3, such leakage can dramatically affect performance. The authors have not provided a fair comparison with other methods, which makes the evaluation here not convincing.
8. In both protein and antibody design tasks, the metrics of perplexity and aar have been misleadingly employed in the field to evaluate protein folding models. The focus of these metrics on "local" recovery rather than entire sequences can inflate performance figures. A more accurate gauge would be to use the AF2 metric to assess structure recovery.

Overall, I commend the authors' ambition to introduce a universal model for protein-related tasks. However, their review of prior works appears incomplete, and the comparisons in their experiments lack rigor. Consequently, this paper does not meet the acceptance standards of ICLR.

[1] Shan et al. “Deep learning guided optimization of human antibody against sars-cov-2 variants with broad neutralization”, PNAS, 2022

[2] Zhang et al. “Protein representation learning by geometric structure pretraining”, ICLR, 2023

[3] Rao et al. "Evaluating protein transfer learning with TAPE." NeurIPS, 2019

[4] Elnaggar et al. “Prottrans: Toward understanding the language of life through self-supervised learning”, PNAS, 2021

[5] Elnaggar et al. “Ankh: Optimized Protein Language Model Unlocks General-Purpose Modelling”, 2023

[6] Madani et al. “Large language models generate functional protein sequences across diverse families”, Nature Biotech, 2023

[7] Wang et al. “Learning Hierarchical Protein Representations via Complete 3D Graph Networks”, ICLR, 2023

[8] Fan et al. “Continuous-Discrete Convolution for Geometry-Sequence Modeling in Proteins”, ICLR, 2023

[9] Jing et al. “Equivariant Graph Neural Networks for 3D Macromolecular Structure”, 2021

[10] Zhang et al. “Pre-Training Protein Encoder via Siamese Sequence-Structure Diffusion Trajectory Prediction”, 2023

[11] Townshend et al. “ATOM3D: Tasks On Molecules in Three Dimensions”, NeurIPS Dataset and Benchmark Track, 2022

[12] Pages et al. “Protein model quality assessment using 3d oriented convolutional neural networks”, Bioinformatics, 2019

[13] Barlow et al. "Flex ddG: Rosetta ensemble-based estimation of changes in protein–protein binding affinity upon mutation." The Journal of Physical Chemistry, 2018

[14] Zhang et al. “Enhancing protein language models with structure-based encoder and pre-training“, 2023

[15] Gligorijevic et al. “Structure-based protein function prediction using graph convolutional networks”, Nature Communications, 2021

---

> ### Author Response · Authors · 2023-11-20
> **Response to Reviewer oPJy (Part1)**
>
> We greatly appreciate the careful review and insightful comments from the reviewer. Below, we provide responses to each comment individually.
>
> > Weaknesses 1: The majority of the components within the proposed method are adaptations from prior studies, which the paper fails to acknowledge.
>
> - The novelty of our work lies in the protein structure encoder, the design of the pretraining method, and the generality of the pretrained model. Our approach is applicable to a wide range of downstream tasks, such as affinity prediction, antibody sequence design, and generation of corresponding CDR structures. These are areas in protein representation learning that have not been extensively explored before.
>   - In reference [1], the initial features of residues are based on the local geometric structure of the Cα atoms, and the atomic coordinate information is considered only within the residues as a supplementary structural feature. In contrast, we construct the position direction between any residues i and j using a spatial position encoder.
>   - Additionally, the distance encoder provides distance information between any backbone atoms. We no longer treat atomic-level coordinate information as an auxiliary property of local residue structure, but instead calculate it across the entire protein structure. The pairwise relationships constructed in this way complement the sequence-based self-attention mechanism, continuously updating the model's parameters to enhance representational capacity and performance. In addition, our pretraining objective also differs slightly from the distance prediction, angle prediction, and dihedral prediction mentioned in [2]. As far as I know, those approaches are conducted at residue level. In our case, we recover the distances between atoms by using residue-level pairwise representations, ensuring that while capturing interactions between residues, we can preserve atomic-level fine-grained information as much as possible.
>
> > Weaknesses 2: The paper's "related work" section is not exhaustive.
>
> We appreciate your valuable feedback and acknowledge the oversight in our "related work" section. We recognize the significance of including recent advancements in protein language models like TAPE, ProtTrans, Ankh, and ProGen2, as well as advancements in protein structure representation such as ProNet and CDConv. In the revised manuscript, we will ensure to provide a comprehensive discussion on the connection between our study and these relevant works to enhance the overall context of our research. Thank you for bringing these important references to our attention.
>
> > Q1: About side-chain details and long-range interactions
>
> - For example, in [7,10], the authors represent protein structures using graphs, where each amino acid is treated as a node and edges are defined based on a cutoff radius. This is a common construction method in graph representation learning. However, it often ignores the interactions between distant residues in inter-chain interactions, as the distance scale differs between intra-chain and inter-chain residues. In contrast, our approach considers both the distance and direction information between any two residues, reducing the impact of the cutoff radius on capturing long-range interactions.
> - As you mentioned, the details of side chains are worth exploring, and we have also taken note of this issue. Building all-atom model of proteins is challenging due to the different types and numbers of atoms in different residue side chains. Moreover, considering the low degrees of freedom in side chain atoms, introducing side chain atoms would increase computational complexity without matching the benefits. In [7], the all-atom model added four torsion angles to the backbone atom model but did not model all side chain atoms explicitly like backbone atoms. Instead, we aggregate the position information of side chain atoms into an additional virtual atom representation. We used the same training settings as the backbone atom pretraining and tested it on downstream tasks. The results showed that incorporating side chain information increased computational complexity without significantly improving the model's predictive performance. We hope to find a more reasonable approach to extend it to all-atom representation in future work.
>
> |         | fold  | super | family | EC   | CATH |
> |---------|-------|-------|--------|------|------|
> | All-atom| 45.5  | 78.3  | **98.2** | **88.7** | 50.7 |
> | ProteiNexus| **47.6** | **79.7** | 98.0 | **88.4** | **53.5** |
>
>
> Regarding Weaknesses 3 and the remaining questions about additional experimental details, we will provide a detailed response in the Part2.

---

> > ### Author Response · Authors · 2023-11-20
> > **Response to Reviewer oPJy (Part2)**
> >
> > > Q3: The model quality assessment, used by the authors for evaluation, has a potential data leakage risk during pre-training.
> >
> > Thank you for highlighting the potential risk of data leakage in the model quality assessment. Due to the distinct objectives and loss functions between pre-training and model quality assessment tasks, we inadvertently overlooked the presence of data overlap between the pre-training dataset and the test dataset. Upon examination, we identified that 6/42 and 3/20 native protein structures in the CASP14 and CASP15 test sets, respectively, had appeared in the pre-training dataset. Due to the extensive time required for a complete retraining, it is not feasible to directly assess the impact of data leakage on the results within the allotted discussion time for this session.
> >  - However, the following experiments are designed to alleviate such concerns to some extent. In Table 6 of Appendix C.1, we present the test results on CASP14 and CASP15 for DeepAccNet(40 times the size of our fine-tuning data) trained with and without pre-training. It is evident that, without the influence of pre-training data, our approach has already achieved state-of-the-art results on 6 out of 8 (3 out of 8) evaluation metrics for the CASP14 (CASP15) test set. Nevertheless, we acknowledge the importance of rigorous scrutiny and additional experiments to address concerns related to data leakage in this task. We will supplement these experimental results in the appendix.
> >
> > > Q4: In the model quality assessment, the authors omit essential baselines.
> >
> > Thank you for your comprehensive consideration. Indeed, it is important to incorporate more representative baselines in the task of model quality assessment. Given the multitude of metrics available for evaluating protein model quality, we opted for the most universally recognized measures, GDT-TS and LDDT. Ornate[12] assesses model quality by fitting CAD scores. Therefore, including Ornate as one of our baselines might not be entirely fair to its approach. We will continue to research and consider other suitable baselines to ensure a comprehensive evaluation and comparison of different methods in the task of model quality assessment.
> >
> > > Q5: For binding affinity prediction, the authors neglect to explain their dataset splits—critical to avoid data leakage.
> >
> > In the prediction of binding affinity, we consider each mutation to be unique, and thus, we employ a random split of the dataset into training, validation, and testing sets with a ratio of 9:1:1. While a cross-validation approach would be more reasonable, we plan to further optimize the experimental results in subsequent iterations.
> >
> > > Q6: About EC and fold classification
> >
> > Thank you for acknowledging the additional outstanding baselines we provided for comparison. While achieving state-of-the-art results on the EC classification task, ProteiNexus is not designed solely for protein structure or function classification, unlike most previous protein representation learning approaches. Our goal is to find a universal paradigm to address a range of protein-related computational problems and extend its applicability to fields like drug discovery that have practical value. Of course, improving the accuracy of classification task predictions and GO predictions is also a goal for our future work. We hope to further demonstrate the versatility of our model.
> >
> > >Q7: For protein design tasks, there are serious data leakage problems due to the presence of test data in pre-training dataset.
> >
> > As described in Appendix 3, we believe that the decline in protein design performance is not solely attributed to data leakage; a substantial portion is due to the uneven distribution of pre-training data. We will conduct extensive experiments to validate this hypothesis.
> >
> > > Q8: Evaluation of protein design and antibody design.
> >
> > In protein and antibody design tasks, I believe that perplexity (PPL) and AAR metrics are appropriate. However, due to a potential lack of clarity in the expression of protein and antibody design metrics in the main text, it may lead to some confusion.
> > - In protein design, the PPL and AAR we evaluate are comprehensive assessments of the entire protein. The purpose of protein design is to predict sequences that can fold into specific three-dimensional structures. Since the residue types corresponding to the protein backbone structure are all unknown, we evaluate the entire sequence as a whole.
> > - However, in antibody design, based on the assumption that the sequences and structures of the antibody framework region are known, we only predict the sequences and structures of CDRs. When assessing the quality of recovering CDR structures, we align the predicted structures with the native structures and calculate the RMSD for the CDRs. Additionally, we will introduce additional structural evaluation metrics such as TM-score and LDDT to provide a more comprehensive assessment of structural recovery.

---

> > > ### Comment · Reviewer_oPJy · 2023-11-21
> > >
> > > >**Q3: Data leakage in MQA**
> > >
> > > Even with different pre-training objectives, the author cannot overlook the data leakage problem. If the methods see the native structures during pre-training, it can learn and memorize some characteristics about its structures, which may lead to overestimated results in downstream tasks. I appreciate the authors’ efforts in alleviating this issue during rebuttal. I would like to see more analysis on that on the final version.
> > >
> > > >**Q4: Missing baseline.**
> > >
> > > As the authors claim that “our model exhibits outstanding performance, establishing
> > > new benchmarks across a range of tasks”, I would not change my opinion if the authors choose to ignore the important baselines I suggest.
> > >
> > > >**Q5: Dataset splits for binding affinity prediction.**
> > >
> > > If the authors just randomly split mutations into training and test sets, then the benchmark in Table 2 is unfair and meaningless. The correct evaluation protocol is to split the dataset by the protein similarity, as done in [1]. Predicting mutations on the same protein is much easier than predicting mutations across different proteins.
> > >
> > > >**Q6: Missing datasets and baselines.**
> > >
> > > Again, as the authors claim that “our model exhibits outstanding performance, establishing
> > > new benchmarks across a range of tasks”, I would not change my opinion if the authors choose to ignore the important datasets and baselines I suggest.
> > >
> > > >**Q7: Missing datasets and baselines.**
> > >
> > > Even if “the decline in protein design performance is not solely attributed to data leakage”, the issue still makes the benchmark in Table 4 invalid.
> > >
> > > >**Q8: Evaluation of protein design and antibody design.**
> > >
> > > The problem of AAR as a metric for protein design is that we do not know “whether larger portion of residues can be recovered, the closest structure it can fold into”. That’s why we need to refold the designed sequence with structure recovery metrics.

---

> > ### Comment · Reviewer_oPJy · 2023-11-21
> >
> > I’d like to thank the authors’ response, but I find many questions I raised have not been addressed during rebuttal. Here is my detailed response:
> > >**Weakness 1: Novelty**
> >
> > The authors claim “The novelty of our work lies in the protein structure encoder, the design of the pretraining method, and the generality of the pretrained model.” I respectfully disagree with that.
> >
> > **Comparison with [1].** During rebuttal, the authors propose that the difference between their architectures and [1] is the pairwise position modeling. However, [1] does model the pairwise positional information through their design of spatial attention. Also, if the authors want to emphasize the novelty of their architecture design over [1], there should be at least some experimental comparison with [1] with the same pre-training setting.
> >
> > **Modeling pairwise distance information between backbone atoms.** There have been tons of works, such as [15], using contact maps to model protein structures, before geometric deep learning was proposed. I don’t think this is the novelty of this paper and clearly the authors have not discussed the relation with these works.
> >
> > It is unclear in the paper how the authors “recover the distances between atoms by using residue-level pairwise representations” during pre-training. Also, the distance prediction methods proposed in [2] can be easily adapted to atom-level as baselines, just as a baseline in [10].
> >
> > >**Q1: About side-chain details and long-range interactions**
> >
> > The authors’ response do not answer my questions.
> >
> > **About long-range interactions.** My question is not about how the method is designed to model long-range interactions, but “the paper lacks experiments substantiating their benefits of long-range interaction”.
> >
> > **About side-chain atom modeling.** I’m not asking for proving the benefits of side-chain atom modeling. My question is: “the paper claims to use atom-level structures, which graph-based methods cannot.” If the author admit that the graph-based methods are able to model side-chain information, they should change their criticism about graph-based methods. Another question is “the authors' method focuses solely on backbone-level structures”. Typically, we only consider models with side-chain information as atom-level encoders. The author should change their claim about the model.
> >
> > Besides, I find the authors miss the discussion with the atom-level GVP work [9] during rebuttal.

---

### Official Review · Reviewer_wGDE · 2023-10-25

**Soundness:** 3 good
**Presentation:** 3 good
**Contribution:** 2 fair
**Rating:** 5
**Confidence:** 4

**Summary:**

This paper proposes a novel pretraining approach for protein representation learning that integrates sequential and structural information. The structural encoding mechanism enables the encoder to learn protein distance information and spatial relative positions of residues, overcoming the inherent drawbacks of ignoring long-range interactions of graph-based representations. As a result, the authors present the model ProteiNexus pretrained by a hybrid masking strategy and mixed-noise strategy to comprehensively capture the structural information. The model is fine-tuned with lightweight task-specific decoders, culminating in exemplary performance across a range of downstream tasks, especially in the understanding of protein complexes.

**Strengths:**

Originality: The paper proposes a novel pretraining strategy which effectively integrates the sequential and structural encoding for the representation learning of proteins. Therefore, the paper uniquely contributes to the field by implementing a simple, yet potent, architecture to capture structural information comprehensively.

Quality: The paper carefully designs the experiments to support the idea. In particular, the extensive experimental results and experimental details presented in the manuscript reflect the comprehensive work of the authors.

Clarity: The paper effectively communicates its ideas and findings with clarity. The paper is well-written, and the logic is coherent. The experimental settings and findings are structured and easy to find the related contents.

Significance: The paper focuses on improving representation learning for proteins as a foundation model for multiple downstream tasks. The model proposed in the manuscript is able to encode sequential and structural data, and surpass baselines on many downstream tasks, illustrating its potential in even more applications in protein design and discovery.

**Weaknesses:**

1. Although the paper is well-written, logically coherent, and self-consistent, I'm afraid the novelty of the paper is not too high. The pertaining strategy which combines sequential and structural information is not new to the field. Furthermore, the major contribution of the paper, which is encoding both the atom-level and the finer-grained distance information, has also been studied extensively in recent years. Therefore, I cannot be persuaded of the novelty of the manuscript unless the authors can provide more evidence about how their model differentiates from existing methods and how their adaptations contribute to the enhanced model performance.

**Questions:**

1. For pretraining experiments, I'm wondering how the noise level is determined. Besides, I'm wondering whether the authors have evaluated the data efficiency of the proposed pertaining approach by varying the pertaining data sizes.

2. For fine-tuning experiments, are the encoder parameters also fine-tuned or frozen? Besides, since the focus of this paper is to evaluate the capability of the encoder and the decoders are already lightweight, why not simply fix the decoder architecture for every downstream task?

3. The authors mention in the conclusion that "... an efficient pre-training model ...", but I'm wondering how the "efficiency" is illustrated: does it enhance model performance, have less computation cost, or require less training data?

---

> ### Author Response · Authors · 2023-11-20
> **Response to Reviewer wGDE (Part1)**
>
> Thank you for the comprehensive comments. I will address your questions one by one and provide insights about novelty.
>
> > Q1: 1. For pretraining experiments, I'm wondering how the noise level is determined. Besides, I'm wondering whether the authors have evaluated the data efficiency of the proposed pertaining approach by varying the pertaining data sizes.
>
>  - We have detailed our chosen noise levels in Table 11 in Appendix D.2. We use different probabilities to mask the proportions of residues and determine the level of noise introduced. Specifically, we employ probabilities of 0.6, 0.2, and 0.2 to mask 15%, 50%, and 100% of sequence lengths, respectively. Additionally, we apply varying degrees of perturbation to the atom coordinates of these masked residues. More specifically, we introduce noise with probabilities of 0.2 and 0.8, following normal distributions with parameters N(0,0.1) and N(0,1), respectively. This introduces noise in the spatial relationships between atoms.
>
>  - Simultaneously, we conducted an in-depth investigation into the impact of different noise levels on pretraining effectiveness. Firstly, we explored uniformly distributed noise within the range of (-1,1). Secondly, we adjusted the proportion of protein residues being masked and increased the probability of completely masking sequence lengths. We applied masking to 15%, 50%, and 100% of sequence lengths with probabilities of 0.3, 0.2, and 0.5, respectively. The results indicate that the strategy of masking residues and the introduced noise level significantly influence pretraining effectiveness. Increasing the difficulty of the pretraining task contributes to enhancing the model's generalization.
>
> |            | fold | super | family | EC  | CATH |
> |------------|-------|-------|--------|------|------|
> | noise-level| 34.1 | 52.7 | 95.4   | 78.0   | 49.6 |
> | mask-level | **50.8** | **81.7** | **98.1** | 87.2 | **55.9** |
> | ProteiNexus| 47.6 | 79.7 | 98.0 | **88.4** | 53.5 |
>
>  - Regarding the evaluation of the impact of pretraining data on downstream tasks, we have not yet experimented with changing the size of the pretraining data. From the results shown in Experiment 6 of Table 13 and Table 14, it is evident that the size of the pretraining data has a significant influence on the results. We will conduct more detailed evaluations by selecting different sizes of pretraining data and applying different levels of redundancy reduction to assess their effects.
>
> > Q2: For fine-tuning experiments, are the encoder parameters also fine-tuned or frozen? Besides, since the focus of this paper is to evaluate the capability of the encoder and the decoders are already lightweight, why not simply fix the decoder architecture for every downstream task?
>
> During the fine-tuning process, we did not freeze the parameters of the encoder.
>  - First, we conducted an experimental comparison on whether to freeze encoder parameters during fine-tuning, and the results are as follows:
>
> |                    | fold | super | family |  EC  | CATH |
> |-------------------- |------|-------|--------|------|------|
> | fixed pretrain model| 35.1 | 58.9  | 96.9   | 84.9 | 52.6 |
> | ProteiNexus         | 47.6 | 79.7  | 98.0   | 88.4 | 53.5 |
>
>  - Although we obtained a universal protein representation through pretraining, when fine-tuning for specific downstream tasks, we preferred to transition from the universal representation to task-specific representations. Additionally, during the fine-tuning stage, we still required the pretrained model's ability to process input data with noise. In binding affinity task, the structures of mutants are often unknown. Most computational methods typically use the wild-type structure as a substitute or predict the mutant structure based on the wild-type using other software. However, these approaches do not accurately represent the native mutant structure. By leveraging the denoising capability of the pretrained model, we can mitigate the impact of such introduced errors. In the task of antibody CDR sequence structure co-design, we can also update the representation of CDR using the pretrained model. This allows us to generate accurate CDR sequences and structures from known framework regions, thereby avoiding the need for an additional decoder and improving computational efficiency.

---

> > ### Author Response · Authors · 2023-11-20
> > **Response to Reviewer wGDE (Part 2)**
> >
> > > Q3: The authors mention in the conclusion that "... an efficient pre-training model ...", but I'm wondering how the "efficiency" is illustrated: does it enhance model performance, have less computation cost, or require less training data?
> >
> > Our efficiency is primarily manifested in several aspects.
> >  - Firstly, we are capable of addressing various protein computation tasks based on a universal pre-trained model, eliminating the need for designing specific model architectures for each particular task.
> >  - Secondly, there is an enhancement in model performance. Our pre-training approach aims to improve the representation learning process, allowing for the elevation of downstream task performance by effectively utilizing protein structure data, even in scenarios where annotated training data is scarce.
> >
> > > Weaknesses: I'm afraid the novelty of the paper is not too high.
> >
> > I consider the novelty of this work lies in the proposed protein structure encoder, pretraining strategy, and the generality of the pretrained model.
> >
> >  - While there have been many works in the field of protein structure representation learning, their adaptability to various tasks is not straightforward. Most of these works excel in protein structure or functional classification tasks, without further exploration in a broader range of tasks. However, our approach demonstrates the ability to extend a simple and effective model to a wide range of tasks, including classification, regression, sequence generation, and structure generation, while consistently achieving promising results.
> >  - Although various protein structure encoders have emerged, adapting to protein complexes remains challenging. By employing a concise yet efficient encoder, we have successfully established the relative positional relationships between residues, aided by distance relations between pairs of backbone atoms. Without the need for intricate modeling, our model vividly presents intra-chain contacts and inter-chain interactions in pair representations. In tasks involving protein-protein interactions, such as binding affinity prediction and antibody design, our model showcases its accurate modeling capabilities for interactions within protein complexes.
> >  - For the pretraining strategy, in addition to the mixed noise, we introduced a novel self-supervised task. Unlike the simple denoising task, we aimed to recover accurate interatomic distances of residue-level pair representations that incorporate information from a hierarchical structure. In a more challenging pretraining environment, the model is forced to learn more extensive and in-depth feature representations.

---

> > > ### Comment · Reviewer_wGDE · 2023-11-22
> > >
> > > Thanks for the author's response to my questions and concerns. I really appreciate that you have helped me better understand your contributions and resolved some of my concerns.
> > >
> > > However, I'm afraid the novelty part is still not convincing: although the authors provide several strengths of their approach, these have already been seen in many other works (also suggested by other reviewers), and the model in this paper does not stand out in terms of the model architecture, training strategies, downstream tasks, etc.
> > >
> > > Therefore, given my concerns which haven't been fully addressed and the criticisms pointed out by other reviewers, I'm sorry that I cannot raise my score. Wish you all the best in the future.

---

### Official Review · Reviewer_tXRC · 2023-10-30

**Soundness:** 2 fair
**Presentation:** 2 fair
**Contribution:** 2 fair
**Rating:** 3
**Confidence:** 4

**Summary:**

The paper introduces a new pretraining approach for learning protein representations, which integrates both structural information and information about downstream tasks. The paper compares their approach to the state-of-the-art on a range of different tasks and report very favourable results.

**Strengths:**

The paper presents a potential solution to a highly relevant problem. The authors have compiled a very comprehensive list of relevant downstream tasks, and thereby make a good the case for a generally useful pretrained model. The reported results are highly competitive. If they hold, the method could thus have a real impact for practitioners in the application domain.

**Weaknesses:**

First of all, I have a slight concern about how well this manuscript fits within the scope of the ICLR venue. Although the title and the introduction point towards a new method, the effective focus of the paper seems to be on benchmarking their method, rather than on the method itself. It seems odd to me that most of the description of the model itself has been moved to the appendix, despite the fact that this would presumably be the most interesting part for most of the ICLR community. For instance, the procedure for fine-tuning on downstream tasks is described in a single sentence in the main text - although it is quite central to their approach.

Secondly, despite the fact that the focus is on the benchmarking, there are details missing about the experiments that makes it difficult for me to judge how much faith can be placed in the reported results. For several experiments (see details below), it is unclear how data was split between training, validation and test - and whether this was done in the same way for all methods that were compared in the result tables. Of particular importance, I did not find it clear whether there was overlap in the pre-training data and the data used for downstream testing. In appendix E.3, the authors have a few reflections on this topic, and seem to show a substantial drop in performance for the protein design task when removing part of the pretraining set. This seems like a red flag to me, which should be investigated more thoroughly - and not only for the protein design task.

Finally, the paper itself does not provide a good explanation for why their approach outperforms prior methods. For the community, it would be useful to know if this relies primarily on the structural signal or the fine-tuning on downstream tasks. These questions are potentially partially addressed by the ablation study in appendix E, but the ablation results are never discussed in the main paper. Also, as far as I could see, the effect on fine-tuning on downstream tasks is not ablated - i.e. the difference between fine-tuning the entire pre-trained model or training a downstream model on a fixed pre-trained model.

**Questions:**

Page 2. *"Predominantly, graph-based representations struggle to preserve ﬁne-grained atom information effectively. Moreover, they tend to accentuate interactions among neighboring residues while often disregarding the inﬂuence of longrange interactions."*
Could you provide a reference to back up this statement?

Page 3. *"Additionally, there are methods that transfer protein structures into distance matrices and attempt to denoise noisy distance matrices while simultaneously predicting the types of corresponding residue types. These approaches undergo pre-training on large-scale datasets to improve the quality and generalizability of the representations."*
Could you provide citations for these methods?

Page 4. *"Lastly, we partition the continuous coordinates into bins of equal width and transform each bin into an embedding"*
Could you describe the motivation for this choice to discretize?

Page 4. *"the "distance tokenizer" method"*
As far as I can see, this method has not been introduced in the paper. Could you elaborate?

Page 4. *"Speciﬁcally, we employ a one-hot encoding scheme to represent the relative distance between position i and position j in the sequence"*
Why use a one-hot encoding to represent a distance? - doesn’t this mean that there is no distinction between bins that are similar in distance and bins that are far apart?

Page 4. *"To better capture the global features and interactions of protein structures, we have opted for the transformer architecture as the backbone of our network. This decision is grounded in the inherent self-attention mechanism of the transformer, which enables computations across the entire protein sequence."*
I don’t understand the distinction you make here. If the graph attention is not capturing enough of the interactions you wish for, can’t you then change the graph to include more interactions? In particular, as far as I can see, graph attention in a fully connected graph would be identical to the attention in a transformer. From that perspective, isn't your approach just a special case of graph attention?

Page 5. *"masked residues"*
"What does “masked residue” mean exactly. Are you masking the identity, or also the atom coordinates?"

Page 5. *"encourage the model to recover authentic atom-level coordinate from noise-induced residue-level pair representations."*
Could you be more precise? How is this "encouraged"?

Page 5. *"Our training dataset includes decoys derived from 7992 unique native protein structures, obtained from DeepAccNet. In the end, we have a collection of 39057 structures in our training dataset, with a fraction representing native structures."*
Since it is central to the data generation process, you should explain in detail what DeepAccNet is, and why it makes sense to use decoys generated by this method as training data. EDIT: I see that you introduce DeepAccNet in the next paragraph, so part of the problem could be resolved by moving that introduction up here. But even when doing so, it is still not clear how the decoys are generated by this method, since it as far as I can see normally produces LDDT scores as output.

Page 5. *"our test set is meticulously curated. It includes targets with experimentally resolved structures from CASP14 and CASP15, paired with their corresponding predicted structures. To ensure diversity and representativeness, we perform a redundancy reduction process on the test set, limiting sequence identity between targets to within 70%."*
Do you also ensure that there is no overlap (high sequence similarity) between the test set and the 7992 structures in your training set?
The choice of homology reduction to 70% seems rather high to me (we generally use values at 30% to avoid leakage). Why was this choice made? I guess you could verify whether this is a problem by plotting the performance as a function of homology to the nearest protein in the training set.

Page 5. *"We validate our pre-training model on ﬁve datasets"*
Does this mean that you in this case do not train a downstream model, but directly use the frequencies of the pretrained model to obtain and estimate of the binding affinity. This should be clarified. If you do use a downstream model, you should clarify how the splits for training/test were constructed (and whether they overlap with the pretraining set). In particular, it is important to establish whether the methods in Table 2 are actually comparable (i.e whether we believe that none of them have been trained on the current test set).

Table 3
Were these other methods run with exactly the same train/test splits as you run your method. In other words, are the results comparable?

Page 7. *"LSTM (Rao et al., 2019), mLSTM (Alley et al., 2019) and CNN Shanehsazzadeh et al. (2020)."*
It us a bit odd that you use architecture names to refer to specific trained models. It would be clearer if you for instance referred to the first as TAPE-LSTM, and the second as the UniRep model.

Page 7. *"ESM-1b"*
The collection of baseline methods was a bit confusing. For instance, you mention language models like ESM-1b and ProtBert-BFD. How are these employed for fold and enzyme-catalyzed reaction classification? Do you somehow use them in an unsupervised way, or do you put a classification layer on top. If so, it would be clearer if you referred to them by a different name than the language model on which they are based.

Table 4.
Again, it is unclear if these methods has been trained and tested on exactly the same datasets - in particular since some of the results are copied from other papers. Please clarify.



### Minor comments:

Page 1. "For instance antibodies (such as SARS-CoV2)"
Rephrase. SARS-CoV2 is not an antibody.

Page 1. *"in protein sequences (Consortium, 2019)"*
Change reference to reflect which consortium

Page 1. *"triumph in various tasks including...protein structure prediction (Rao et al., 2020;"*
This paper is about contact prediction, not directly about protein structure prediction.

Table 3. Caption. The title currently says *"Results of classification"*. Would be helpful if you could specify the experiment in the title.

---

> ### Author Response · Authors · 2023-11-21
> **Response to Reviewer tXRC (Part1/3)**
>
> Thank you for your thorough review and valuable suggestions. We appreciate the insightful comments you provided, and we have carefully considered each one. Your feedback has been instrumental in improving the quality of our work.
>
> > Weaknesses 1: First of all, I have a slight concern about how well this manuscript fits within the scope of the ICLR venue.
>
> Thank you for your review and valuable feedback. We understand your concerns regarding the limited description of the method itself in the main text, and we will make adjustments in the final version to better highlight the method we propose. Our structure-based pretraining is aimed at discovering general patterns for solving various protein computing tasks, and the extensive testing across downstream tasks is intended to demonstrate the effectiveness and applicability of the model. Regarding the description of the fine-tuning process you mentioned, we do recognize it as the core of our method, and we will further elaborate on this aspect in the main text to present the key steps of our method more clearly. With these adjustments, we hope to better meet the expectations of the ICLR community for a comprehensive description of the model method itself.
>
> > Weaknesses 2: Secondly, despite the fact that the focus is on the benchmarking, there are details missing about the experiments that makes it difficult for me to judge how much faith can be placed in the reported results.
>
> Regarding the issue of dataset split, we will respond to each specific question below. Regarding the potential data leakage between pretraining data and downstream task fine-tuning data, we would like to make the following points:
>  - Firstly, tasks susceptible to data leakage are limited to model quality assessment and protein design. For other tasks such as binding affinity prediction, folding classification, etc., not only is additional annotated data introduced, but also the protein sequence and structural information serve as known conditions for the tasks. Hence, there is no conflict between the self-supervised pretraining tasks and downstream tasks, avoiding the impact of data leakage.
>  - Secondly, concerning the data leakage issue in protein design, we have provided a brief explanation in Appendix E.3. We believe that the distribution of pretraining data significantly influences protein design outcomes. Therefore, we plan to cluster the pretraining data based on redundancy and elaborate on the actual reasons leading to the decrease in model performance. For the potential data leakage issue in model quality assessment, you can refer to the explanation provided in response to Reviewer oPJy (Part 2) Question 3.
>
> > Weaknesses 3: Finally, the paper itself does not provide a good explanation for why their approach outperforms prior methods.
>
> During the fine-tuning stage, we optimize the parameters of the pre-trained model simultaneously. Freezing the parameters of the pre-trained model may lead to a certain degree of performance decline. Our approach relies on both structural signals and fine-tuning on downstream tasks. As evident from the experimental results in Appendix E, our model functions as an organic whole. The effective integration of structural information, modeling of backbone atoms, and the choice of pre-training data and strategies contribute to our competitive performance across various tasks. While our pre-trained model extracts universal protein representations, different downstream tasks exhibit varying preferences for the information contained in these representations. We conducted experiments by training downstream models on a fixed pre-trained model, and the results are as follows:
>
> |                     | fold   | super  | family | EC    | CATH  |
> |---------------------|--------|--------|--------|-------|-------|
> | fixed pretrain model | 35.1   | 58.9   | 96.9   | 84.9  | 52.6  |
> | ProteiNexus     | **47.6** | **79.7** | **98.0** | **88.4** | **53.5** |

---

> > ### Author Response · Authors · 2023-11-21
> > **Response to Reviewer tXRC (Part 2/3)**
> >
> > > Q1: Could you provide a reference to back up this statement?
> >
> > Our perspective is supported by recent literature. Recent studies have highlighted the issue of "over-squashing" in Graph Neural Networks (GNNs) when propagating information from distant nodes. This phenomenon is attributed to bottlenecks in the graph, where the rapid increase in the number of distant neighbors leads to excessive compression of information. In support of our assertion, relevant literature [1,2] provides in-depth analyses of the over-squashing issue and proposes various solutions, including curvature-based graph rewiring methods. Our viewpoint is grounded in current research on bottleneck problems in Graph Neural Networks, which may hinder the efficiency of GNNs in handling interactions with distant nodes.
> >
> > [1] Alon et al. “On the Bottleneck of Graph Neural Networks and its Practical Implications”, ICLR, 2021
> >
> > [2] Topping et al. “Understanding over-squashing and bottlenecks on graphs via curvature”, ICLR, 2022
> >
> > > Q2: Could you provide citations for these methods?
> >
> > [3] Zhou et al. “Uni-mol: A universal 3d molecular representation learning framework”, ICLR, 2023
> >
> > > Q3: Could you describe the motivation for this choice to discretize?
> >
> > Discretizing spatial structural information, such as distance, coordinates, angles, etc., is the simplest, direct, and effective method for encoding structural information. This embedding provides a low-dimensional yet rich representation of the original continuous coordinates, reducing the computational burden associated with continuous values.
> >
> > > Q4: Page 4. "the "distance tokenizer" method" As far as I can see, this method has not been introduced in the paper. Could you elaborate?
> >
> > Similar to natural language, we represent distances as a sequence of discrete tokens obtained from a "distance tokenizer," rather than using the raw distance values. We discretize the distance between residue i and j into |v| bins, covering the range from 0 to 127 Å, with equal-width bins except for the last one, which includes any more distant residue pairs. To facilitate hierarchical learning of distance representations at different levels of precision, we discretize distances into bins of varying lengths, where |v| = {16, 64, ... 16384}, and then aggregate them into the initial atom-level distance encoding. Additional details can be found in Appendix B.1.
> >
> > > Q5: Why use a one-hot encoding to represent a distance? - doesn’t this mean that there is no distinction between bins that are similar in distance and bins that are far apart?
> >
> > From a sequence perspective, residue pairs with close and distant distances are placed in entirely different bins, representing their distinct nature. More precisely, for a residue pair (i, j), where i and j range from 1 to the sequence length N, we calculate the clipped relative distance within a chain and encode it as a one-hot vector.
> >
> > > Q6: I don’t understand the distinction you make here. If the graph attention is not capturing enough of the interactions you wish for, can’t you then change the graph to include more interactions? In particular, as far as I can see, graph attention in a fully connected graph would be identical to the attention in a transformer. From that perspective, isn't your approach just a special case of graph attention?
> >
> > Your understanding is absolutely correct. Graph attention in a fully connected graph is equivalent to the attention mechanism in transformer. We chose the transformer primarily because, as mentioned in A1, it can preserve fine-grained atom information and capture long-range interactions effectively.
> >
> > > Q7: Page 5. "masked residues" "What does “masked residue” mean exactly. Are you masking the identity, or also the atom coordinates?"
> >
> > "Masked residues" refer to a portion of residues whose amino acid types are replaced by a 'mask' during the training process.
> >
> > > Q8: Could you be more precise? How is this "encouraged"?
> >
> > Our pre-training process involves two self-supervised tasks: first, the restoration of residue types marked as "MASK," and second, the recovery of residue-level pair representations with noise to atom distances. In the pre-training phase, certain residues are presented in the form of "MASK," and the model, through sequence context information and communication with pair representations, attempts to reconstruct the types of residues marked as 'MASK'. Simultaneously, we encourage the model to learn 3D structural information during pre-training, incorporating atom-level positional information in residue-level representations. The core idea of this task is to perturb the three-dimensional coordinates of input atoms to varying degrees, restoring spatial positional information with noise to accurate atom-level distance relationships.

---

> > > ### Author Response · Authors · 2023-11-21
> > > **Response to Reviewer tXRC (Part 3/3)**
> > >
> > > > Q9: It is still not clear how the decoys are generated by DeepAccNet.
> > >
> > > You have a correct understanding of DeepAccNet, and it is indeed an excellent method for predicting LDDT scores. The confusion may have arisen from our previous statement. The DeepAccNet we refer to here refers to the data set used by the method during training, where decoys were generated using comparative modeling with RosettaCM and local structure perturbations.
> > >
> > > > Q10: Do you also ensure that there is no overlap between the test set and the 7992 structures in your training set?
> > >
> > >  - We ensure that there is no overlap between the test set and the 7992 structures in the training set by considering the release dates of the structural data. The crystal structures corresponding to the decoys from the training set were sourced from before May 1, 2018, while structures from CASP14 and CASP15 were released after August 2020.
> > >  - We typically use a 30% similarity threshold to prevent data leakage. However, in this context, our purpose in comparing sequence similarities is to eliminate redundant data in the test set, rather than to avoid data leakage between the training and test sets.
> > >  - The targets in CASP are meant to be representative, and each target requires predictions from the groups participating in the model quality assessment track. Ideally, we would like to retain all test targets. However, to address the specific situation mentioned in Appendix C.1, where H1044 appears as a challenging target in CASP14, we adopted a higher similarity threshold (70%) as the cutoff for redundancy. Given that the H1044 sequence is relatively long, it is partitioned into multiple structural domains (such as T1031, T1033), treated as independent prediction targets. Repeated inclusion of these targets in the test set could lead to biased test results.
> > >
> > > > Q11: You should clarify how the splits for training/test were constructed.
> > >
> > > We retrained the model for predicting binding affinity using a random split of 8:1:1 for training, validation, and testing. Considering the reviewer oPJy's suggestion for cross-validation on a small dataset, we acknowledge the current lack of rigor in our data split and plan to address this issue. Additionally, we believe that any overlap between the data used for binding affinity fine-tuning and pretraining does not affect the results. When predicting changes in protein binding affinity, the protein structure data is known and can reasonably be included as part of the pretraining data. Furthermore, the labels describing changes in binding affinity did not appear during the pretraining process.
> > >
> > > > Q12: Table 3 Were these other methods run with exactly the same train/test splits as you run your method.
> > >
> > > The training, validation, and test sets used in our two classification tasks, as well as the methods mentioned in Table 3, are exactly the same. However, it is worth noting that the first column, including three traditional structural alignment algorithms (TMalign) and sequence alignment algorithms (HHSuite, PSI-BLAST), does not involve a training process. Although we used the same dataset for fine-tuning, the methods described in the "w/pretraining" section utilized different pre-training datasets, as detailed in Section 4.3 "Baselines."
> > >
> > > > Q13: It us a bit odd that you use architecture names to refer to specific trained models.
> > >
> > > To facilitate comparison with previous work, we maintained the naming conventions for different methods in the baseline, which may have caused some confusion in the network structure naming. We will address this issue and appreciate your attention to these details.
> > >
> > > > Q14: "ESM-1b" The collection of baseline methods was a bit confusing.
> > >
> > > ESM-1b and ProtBert-BFD language models were fine-tuned using an MLP prediction head, and the results in Table 3 are from the Multiview Contrast method.
> > >
> > > > Q15: Table 4. Again, it is unclear if these methods has been trained and tested on exactly the same datasets.
> > >
> > > The data split used for fine-tuning in protein design is identical to that of other methods in Table 4. Some subtle differences have been described in Section 4.4's **Baselines**.
> > >  - On the CATH dataset, our division into training, validation, and test sets is entirely consistent with other methods, with the only exception being ESM-IF's use of CATH v4.3, while other methods utilize CATH v4.2.
> > >  - Additionally, methods in the first column of the table did not explicitly specify the training and validation sets used for testing on TS50. In contrast, methods in the second column clearly stated that they utilized data with high consistency removed from the CATH training set with TS50 sequences (referred to as CATH canonical training set for TS50). These methods retrained the model using this data and subsequently tested on TS50. To provide a comprehensive comparison, we present the test results on TS50 using different training sets at the bottom of the table, namely, ProteiNexus (canonical) and ProteiNexus.

---

> > > > ### Comment · Reviewer_tXRC · 2023-11-22
> > > > **Response to rebuttal**
> > > >
> > > > Thanks to the authors for their rebuttal. While the authors make reasonable arguments, they do not seem to have updated their submitted PDF, and only promise to make changes in a "final version". Since several of the necessary modifications are substantial changes to the structure of the paper, not having these available makes it difficult to judge the quality of the promised update. Also, for the many requests for minor changes the authors provide a response in the rebuttal but do not state how they will modify the main text to clarify these issues in the paper itself - which is ultimately what matters.
> > > >
> > > > For these reasons, I will not change my score of this paper at this time, but encourage the authors to resubmit a new version to a future venue.

---

### Meta-Review · Area_Chair_r18c · 2023-12-06

**Metareview:**

The paper considers the important problem of protein representation learning and proposes a pre-training and fine-tuning framework integrating sequence-based and structural information, with the goal of achieving great performance across diverse tasks.

The AC and reviewers find the approach promising. The author feedback and additional experimental results are very much appreciated and we strongly urge the authors to revise their manuscript accordingly. To convincingly demonstrate the value of the approach it is critical to follow up on the promise to evaluate against more recent powerful baselines listed by reviewer oPJy, to correct the data splitting protocol for binding affinity prediction, and to perform additional analysis on data leakage.

**Justification For Why Not Higher Score:**

Many important concerns have not been resolved. Significant revisions,  additional experiments and protocol correction are needed.

**Justification For Why Not Lower Score:**

N/A

---

### Decision · Program_Chairs · 2024-01-16

Reject